# Intuitive physical reasoning about objects' masses transfers to a visuomotor decision task consistent with Newtonian physics

**Nils Neupärtl**[1,2]*, **Fabian Tatai**[1,2], **Constantin A. Rothkopf**[1,2,3]

**1** Centre for Cognitive Science, Technical University of Darmstadt, 64283 Darmstadt, Germany, **2** Institute of Psychology, Technical University of Darmstadt, 64283 Darmstadt, Germany, **3** Frankfurt Institute for Advanced Studies, Goethe University, 60438 Frankfurt, Germany

* neupaertl@psychologie.tu-darmstadt.de

**Data Availability Statement:** All human data are available on Github: https://github.com/RothkopfLab/ploscompbio_pucks/.

## Abstract

While interacting with objects during every-day activities, e.g. when sliding a glass on a counter top, people obtain constant feedback whether they are acting in accordance with physical laws. However, classical research on intuitive physics has revealed that people's judgements systematically deviate from predictions of Newtonian physics. Recent research has explained at least some of these deviations not as consequence of misconceptions about physics but instead as the consequence of the probabilistic interaction between inevitable perceptual uncertainties and prior beliefs. How intuitive physical reasoning relates to visuomotor actions is much less known. Here, we present an experiment in which participants had to slide pucks under the influence of naturalistic friction in a simulated virtual environment. The puck was controlled by the duration of a button press, which needed to be scaled linearly with the puck's mass and with the square-root of initial distance to reach a target. Over four phases of the experiment, uncertainties were manipulated by altering the availability of sensory feedback and providing different degrees of knowledge about the physical properties of pucks. A hierarchical Bayesian model of the visuomotor interaction task incorporating perceptual uncertainty and press-time variability found substantial evidence that subjects adjusted their button-presses so that the sliding was in accordance with Newtonian physics. After observing collisions between pucks, which were analyzed with a hierarchical Bayesian model of the perceptual observation task, subjects transferred the relative masses inferred perceptually to adjust subsequent sliding actions. Crucial in the modeling was the inclusion of a cost function, which quantitatively captures participants' implicit sensitivity to errors due to their motor variability. Taken together, in the present experiment we find evidence that our participants transferred their intuitive physical reasoning to a subsequent visuomotor control task consistent with Newtonian physics and weighed potential outcomes with a cost functions based on their knowledge about their own variability.

**Funding:** The authors received no specific funding for this work.

**Competing interests:** The authors have declared that no competing interests exist.

## Author Summary

During our daily lives we interact with objects around us governed by Newtonian physics. While people are known to show multiple systematic errors when reasoning about Newtonian physics, recent research has provided evidence that some of these failures can be attributed to perceptual uncertainties and partial knowledge about object properties. Here, we carried out an experiment to investigate whether people transfer their intuitive physical reasoning to how they interact with objects. Using a simulated virtual environment in which participants had to slide different pucks into a target region by the length of a button press, we found evidence that they could do so in accordance with the underlying physical laws. Moreover, our participants watched movies of colliding pucks and subsequently transferred their beliefs about the relative masses of the observed pucks to the sliding task. Remarkably, this transfer was consistent with Newtonian physics and could well be explained by a computational model that takes participants' perceptual uncertainty, action variability, and preferences into account.

## Introduction

Whether sliding a glass containing a beverage on a counter top in your kitchen or shooting a stone on a sheet of ice in curling, acting successfully in the world needs to take physical relationships into account. While humans intuitively sense an understanding of the lawful relationships governing our surroundings, research has disputed that this is indeed the case [1, 2]. Instead, human judgements and predictions about the dynamics of objects deviate systematically from the laws of Newtonian mechanics. Past research has interpreted these misjudgments as evidence that human judgements violate the laws of physics and that they instead use context specific rules of thumb, so called heuristics [2–4]. E.g., when judging relative masses of objects such as billiard balls based on observed collisions, people seem to use different features of motion in different contexts and end up with erroneous predictions [2].

But recent research has provided a different explanation of human misjudgments on the basis of the fact that inferences in general involve sensory uncertainties and ambiguities, both in perceptual judgements [5, 6] as well as in reasoning and decision making [7, 8]. Therefore, physical reasoning needs to combine uncertain sensory evidence with prior beliefs about physical relationships to reach predictions or judgements [9–13]. By probabilistically combining prior beliefs and uncertain observations, a posterior probability about the unobserved physical quantities is obtained. Judgements and predictions are then modeled as based on these probabilistic inferences. Thus, deviations from the predictions of Newtonian physics in this framework are attributed to perceptual and model uncertainties.

This framework of explaining reasoning about physical systems on the basis of Newtonian mechanics and perceptual uncertainties has been referred to as the noisy Newton framework (see e.g. [14] for a review). It has been quite successful at explaining a range of discrepancies between predictions of Newtonian physics and human predictions for various perceptual inference tasks, including subjects' biases in judgements of mass ratios when observing simulated collisions of objects, if perceptual uncertainties are taken into account [9, 15]. Additionally, the noisy Newton framework can also explain why human judgements depend on experimental paradigms, because tasks differ in the availability of knowledge about objects' properties [16]. As an example, this suggests an explanation for the fact that judgements about physical situations based on a static image representing a situation at a single timepoint have usually been reported to deviate more from physical ground truth compared to richly

animated stimuli [17], which additionally allow to estimate objects' velocities. Nevertheless, some persistent failures of intuitive physical reasoning have been suggested to be caused by distinct systems of reasoning compared to the more calibrated physical reasoning underlying visuomotor tasks [16].

While physical reasoning has been studied predominantly using tasks in which subjects needed to judge physical quantities or predict how objects continue to move, much less is known about how intuitive physical reasoning guides actions. Commonly, experimental paradigms have asked subjects to judge physical properties in forced choice paradigms such as relative masses in two-body collisions [3, 9, 11, 12], predict the future trajectory of an object when no action is taken based on an image of a situation at a single timepoint, such as a pendulum [16], a falling object [18], or whether an arrangement of blocks is stable [12]. Other experiments have asked subjects to predict a trajectory of objects [19] or their landing position [10] after seeing an image sequence, but again without subjects interacting with the objects in the scene. Recent studies have also investigated more complex inference problems in which subjects needed to learn multiple physical quantities by observing objects' dynamics [13] or quantified how much entropy reduction for forced choice questions about physical properties of objects was achieved by interactions with objects in a scene [20]. By contrast, the literature on visuomotor decisions and control [21–24] has seldom investigated the relationship between visuomotor decisions, actions, and control and physical reasoning. Notable exceptions are studies which have investigated how humans use internal models of gravity in the interception of moving targets [25] and how exposure to 0-gravity environments [26] changes this internal model. Nevertheless, these studies did not investigate the inference and reasoning of unobservable physical quantities. Other studies have investigated how perceptual judgements and visuomotor control in picking up and holding objects in the size-weight and material-weight illusions can be dissociated [27, 28]. Nevertheless, these studies did not investigate the relationship of intuitive physical reasoning and visuomotor actions.

Here we investigate how human subjects guide their actions based on their beliefs about physical quantities given prior assumptions and perceptual observations. Thus, we combine work on intuitive physics [9, 11, 12] and visuomotor control [21, 23, 25, 27]. First, do humans use the functional relationships between physical quantities as prescribed by Newtonian mechanics in new task situations? Specifically, when sliding an object on a surface the velocity with which the object needs to be released needs to scale linearly with the object's mass but with the square-root of the distance the object needs to travel. Second, when interacting with simulated physical objects, do humans interpret differences in objects' behavior in accordance with physical laws? Specifically, when two objects slide according to two different non-linear relationships, subjects may attribute these differences to the lawful influences of unobserved physical quantities such as mass. Third, after having observed collisions between objects do humans adjust their actions to be consistent with the inferred relative masses of those objects? Specifically, while it is known that subjects can judge mass ratios of two objects when observing their collisions, it is unclear whether they subsequently use this knowledge when sliding those objects. To address these questions, subjects were asked to shoot objects gliding on a surface under the influence of friction to hit a target's bullseye in a simulated virtual environment. The simulated puck was accelerated by subjects' button presses such that the duration of a button press was proportional to the puck's release velocity. A succession of four phases investigated what prior assumptions subjects had about the relationships between their actions and physical quantities, whether they could learn to adjust their actions to different objects when visual feedback about their actions was available, whether they would interpret the differences in objects' behavior in accordance with physical laws, and whether they could transfer mass ratios inferred from observing collisions to adjust their actions accordingly.

Analysis of the data shows that subjects adjusted their press-times depending on the distance the pucks had to travel. Furthermore, subjects adjusted the button press-times to get closer to the target within a few trials when visual feedback about the puck's motion was available. Because perceptual uncertainties and motor variability can vary substantially across subjects and to take Weber-Fechner scaling into account, we subsequently analyzed the data with a hierarchical Bayesian interaction model under the assumption that subjects used a Newtonian physics based model. We compared this model to the prediction of a linear heuristics model. Importantly, because subjects needed to adjust their button press-times, the model needs to account for perceptual judgements and the selection of appropriate actions. We include a comparison of three cost functions to investigate subjects' selection of press-times. Based on this model of the sliding task, we find evidence that subjects used the functional relationship between mass and distance of pucks as prescribed by Newtonian physics and readily interpreted differences between two pucks' dynamics as stemming from their unobserved mass. Moreover, biases in subjects' press-times can be explained as stemming from costs for not hitting the target, which grow quadratically with the distance of the puck to the target's bullseye. After observing 24 collisions between an unknown puck and two pucks with which subjects had previously interacted, we found evidence that participants transferred the inferred relative masses to subsequent sliding actions. The mass beliefs from observing the collisions were inferred by a hierarchical Bayesian observation model. Thus, intuitive physical reasoning transfers from perceptual judgements to control tasks and deviations from the predictions of Newtonian physics are not only attributable to perceptual and model uncertainties but also to subjects' implicit costs for behavioral errors.

## Materials and methods

### Participants

Twenty subjects took part in the experiment. All participants were undergraduate or graduate students recruited at the Technical University of Darmstadt, who received course credit for participation. All experimental procedures were carried out in accordance with the guidelines of the German Psychological Society and approved by the ethics committee of the Technical University of Darmstadt. Informed consent was obtained from all participants prior to carrying out the experiment. All subjects had normal or corrected to normal vision and were seated so that their eyes were approximately 40 cm away from the display and the monitor subtended 66 degrees of visual angle horizontally and 41 degrees vertically. In the vertical direction the monitor had a resolution of 1080 pixels, which corresponded to a distance of approximately $11.5m$ in the simulation. Four participants have been excluded from the analysis (three due to incorrect task execution and one due to incomplete data; f = 9, m = 11, age = [18, 27], median = 22.5, mean = 22.25).

### Experimental design and data

Participants were instructed to shoot a puck in a virtual environment into the bullseye of a target, similar to an athlete in curling. The shot was controlled by the duration of pressing a button on a keyboard. Participants were told that they were able to adjust the force, which initially was going to accelerate the puck and thus the initial velocity of the puck, by the duration of their press. However, they were not explicitly told about the linear relationship between the press time and the initial velocity. Additionally, participants were told that realistic friction was going to slow down the puck while sliding on the simulated surface. The general objective of the experimental design was to investigate whether subjects adjusted their shooting of the pucks in a way that was in line with the physical laws governing motion under friction.

Specifically, the magnitude of the initial impulse exerted on the puck determines how far the puck slides on the surface. Thus, subjects needed to adjust the duration of a button press according to the distance between the randomly chosen initial position of the puck and the target on each trial. The different experimental phases allowed investigating subjects' prior beliefs about the puck's dynamics, their adjustments of button presses when these beliefs were updated given visual feedback of the puck's motion, and the potential transfer of knowledge about relevant object properties to the control of the puck from perceiving object collisions. Therefore we designed a task with two conditions and four consecutive experimental phases, which differed in the availability of previous knowledge and feedback.

**Laws of motion governing the puck's motion.**   At the beginning of each trial, subjects saw the fixed target and a puck resting at a distance chosen uniformly at random between one and five meters from the target's bullseye. To propel the puck toward the target, subjects needed to press a button. To model the relationship between the button press and the puck's motion, we reasoned as follows. Human subjects have been shown to be able to reason accurately about the mass ratio of two objects when observing elastic collisions between them [9]. In elastic collisions, according to Newtonian laws, the impulse transferred by the collision is proportional to the interaction duration with a constant force. In other words, the duration of the interaction with a constant force leads to a linearly scaled impulse. Given a constant mass $m$ of a puck and assuming a constant surface friction coefficient $\mu$, Newtonian physics allows deriving the button press-time $T_{press}$ required to propel the puck to the target at a distance $\Delta x$:

$$T_{press} = \sqrt{\frac{2\mu g m^2}{F^2}\Delta x} \propto m \cdot \sqrt{\Delta x} \tag{1}$$

with gravitational acceleration $g$ and a constant force $F$. Here, the constant force $F$ is being applied by the interaction, i.e. the button press of duration $T_{press}$, which is physically equivalent to an elastic collision with an object. Note that this formulation of the interaction has the additionally intuitive consequence that the release velocity of the puck scales linearly with the duration of the button press (see S1 Appendix "Puck Movement"). The second expression clarifies that the press-time scales linearly with the mass of the puck, while it scales with the square-root of the distance to the target. Obviously, this relationship assumes perfect knowledge of all involved quantities. The movement of the puck was implemented by simulating the equivalent difference equations for each frame given the friction and the velocity of the preceding frame (detailed derivations are provided in the S1 Appendix, "Puck Movement").

**Phase 1: Prior beliefs.**   In the first phase, we wanted to investigate, which functional relationship subjects would use a priori to select the duration of button presses depending on the perceived distance between the puck and the target. A black puck with unknown mass $m$ was placed at a distance to the target drawn uniformly at random. Participants received no further information about the puck or the environment. Participants were instructed to press the button in a way so as to bring the puck into the target area, but after pressing the button for a duration $t^{pre}$ and releasing it the screen turned black to mask the resulting movement of the puck. This screen lasted for at least half a second until the participant started the next trial by button press. All participants carried out fifty trials. Thus, the collected data allowed relating different initial puck distances to the press-times subjects selected based on their prior beliefs.

**Phase 2: Visual feedback.**   The second phase was designed to investigate how participants adjusted their button press-times in relation to the simulated masses of pucks and their initial distances to the target when visual feedback about the pucks' motion was available. To this end, participants carried out the same puck-shooting task but with two different pucks, as indicated by distinct surface textures (yellow diamond versus five red dots, see Fig 1b, *Feedback*).

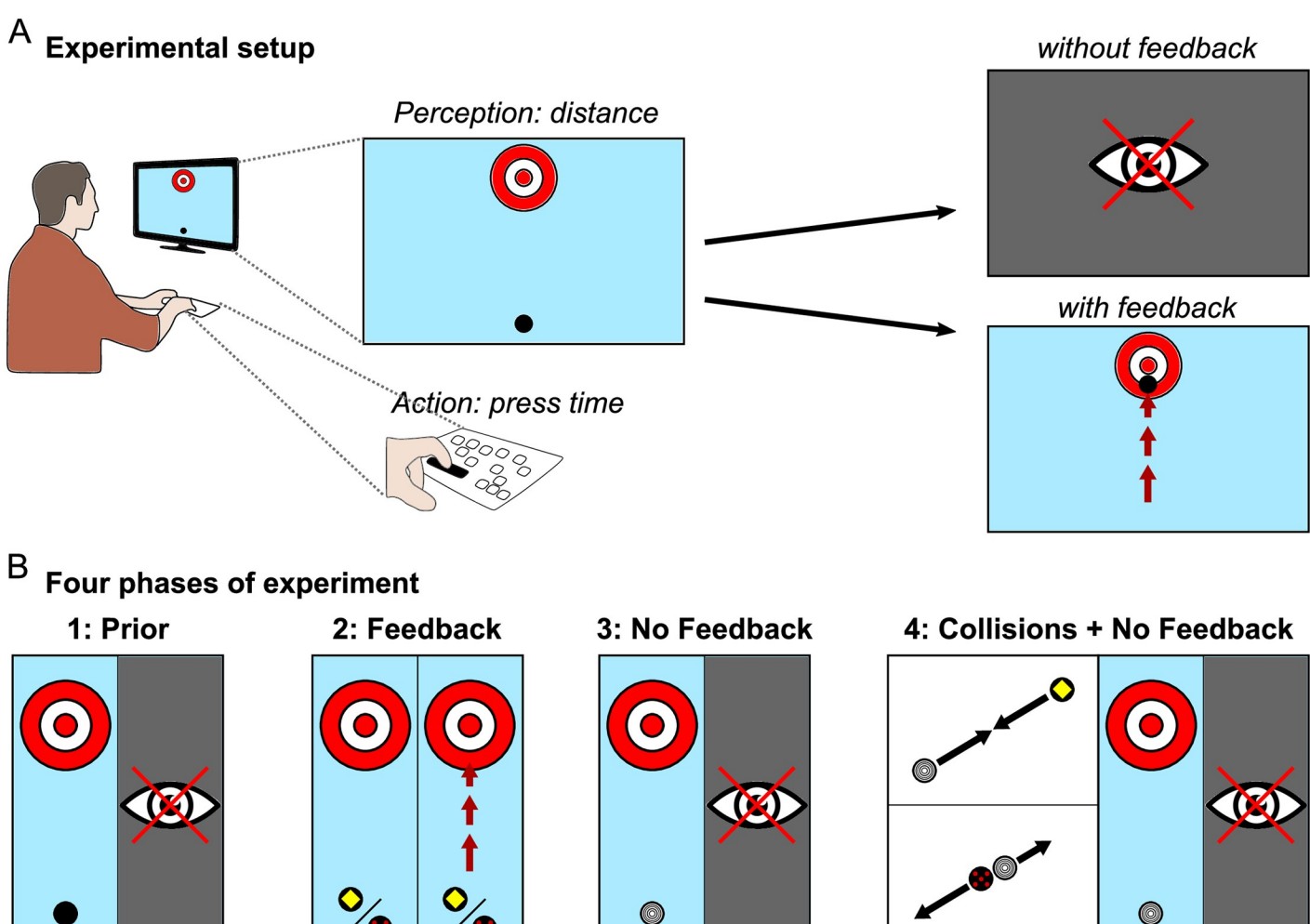

**Fig 1. Task design. (A)** Single trial illustration. Target area and puck are presented on a monitor from bird's-eye perspective. Releasing the pressed button accelerates the puck by applying a force, which is proportional to the press-time. In trials without feedback the screen turned black after button release, while in feedback trials participants were able to see the puck moving according to simulated physics. **(B)** Four phases of the experiment. In the 'prior' phase, no feedback about puck motion was available, whereas in the 'feedback' phase subjects obtained visual feedback about the pucks' motion. Two pucks with different colors and correspondingly different masses were simulated. In the 'no feedback' phase subjects obtained a new puck as indicated by a new color and obtained no feedback. In the last phase, subjects first watched 24 collisions between the new puck and the pucks they had interacted with in the 'feedback' phase before interacting again with the puck. Note that the puck of the 'no feedback' and 'collisions + no feedback' phase are identical.

The two pucks were alternating every four trials with a total number of two-hundred trials. The two different pucks were simulated with having differing masses, resulting in different gliding dynamics. In this condition, participants received visual feedback about their actions as the pucks were shown gliding on the surface from the initial position to the final position depending on the exerted impulse. Thus, because the distances traveled by the two pucks for different initial positions as a function of the button press-times $t^{pre}$ could be observed, participants could potentially use this feedback to adjust their press-times on subsequent trials. Note that the two pucks were only distinguished by a color cue and no cue about mass was given apart from the different dynamics. Half the participants were randomly assigned to the 'light-to-heavy' condition, in which the two pucks had masses of 1.5 kg and 2.0 kg, and the other half of the participants were assigned to the 'heavy-to-light' condition, in which the pucks had masses of 2.0 kg and 2.5 kg.

**Phase 3: No feedback.** In phase three, we wanted to investigate how having observed the sliding of the pucks in phase two influenced participants' press-times with an unknown puck. Subjects were asked to shoot a new puck they had not seen before to the target without visual feedback, as in the first experimental phase, for one-hundred trials (Fig 1B, *No Feedback*). The texture of the puck consisted of five concentric rings. For participants in the 'light-to-heavy' condition, the new puck had a mass of 2.5 kg whereas for participants in the 'heavy-to-light' condition the new puck had a mass of 1.5 kg. However, different from phase one, in which subjects had not obtained feedback about the pucks' motion, by phase three participants had already interacted with three pucks and obtained visual feedback about the motion of two pucks. Importantly, participants had received feedback about the non-linear nature of gliding under friction in phase two, albeit scaled differently for the two pucks. Thus, this experimental phase allowed investigating, whether subjects use the functional mapping from puck distances to press-times prescribed by Newtonian physics and what assumptions about the mass of an unknown puck they used.

**Phase 4: Collisions & no feedback.** With the final experimental phase we wanted to investigate, whether participants can use the relative mass ratios inferred from observing collisions between two pucks to adjust their subsequent actions with one of those pucks. At the beginning of phase four, participants watched a movie of twenty-four collisions between two pucks. One was always the puck with unknown mass used in phase three (without feedback; five rings) (see Fig 1B, *Collisions No Feedback*), while the second puck was one of the two pucks presented in phase two (see Fig 1B, *Feedback*). Each collision thus showed one of the two previously seen pucks from phase two selected at random colliding with the puck from phase three with a total of twelve collision with each of the two known puck. By observing these elastic collisions participants were expected to learn the mass ratios between pucks, as shown in previous research [9, 15]. Note that the pucks were simulated without the influence of friction in these collisions, ensuring that participants only obtained a cue about relative masses and not about the dynamics under friction for the puck from phase 3. After watching these collisions, subjects were asked to shoot the puck from phase three again without obtaining visual feedback, as in phases one and three, for one-hundred trials. Thus, subjects interacted with the same puck as in phase three but had now seen the collisions of this puck with the two pucks they had interacted with. This experimental phase therefore allowed investigating, whether subjects used the learned mass ratios and transferred them to the control task to adjust their press-times. Importantly, having learned the mass ratios between pucks needs to be transferred to the press-times, which differ in a physically lawful way depending on the initial distance of the pucks to the target. As the two pucks from phase two of the experiment were only distinguished by color, such a transfer indicates that subjects had attributed the different dynamics to their masses consistent with Newtonian physics. Thus, if subjects used an internal model of physical relationships, they should be able to adjust their press-times for the new puck without ever having seen it glide.

## Results

### Behavioral results

As subjects did not receive visual feedback about the consequences of their button presses in the first phase of the experiment, the button press-times reflect the prior assumptions they brought to the experiment. Indeed, subjects' press-times $t^{pre}$ grew with the initial distance between the puck and the target. The button press times for all phases of the experiment are shown in Fig 2. The correlation between $t^{pre}$ and the initial distance was 0.482 ($p < 0.001$). However, the functional relationship according to Newtonian physics prescribes a scaling of

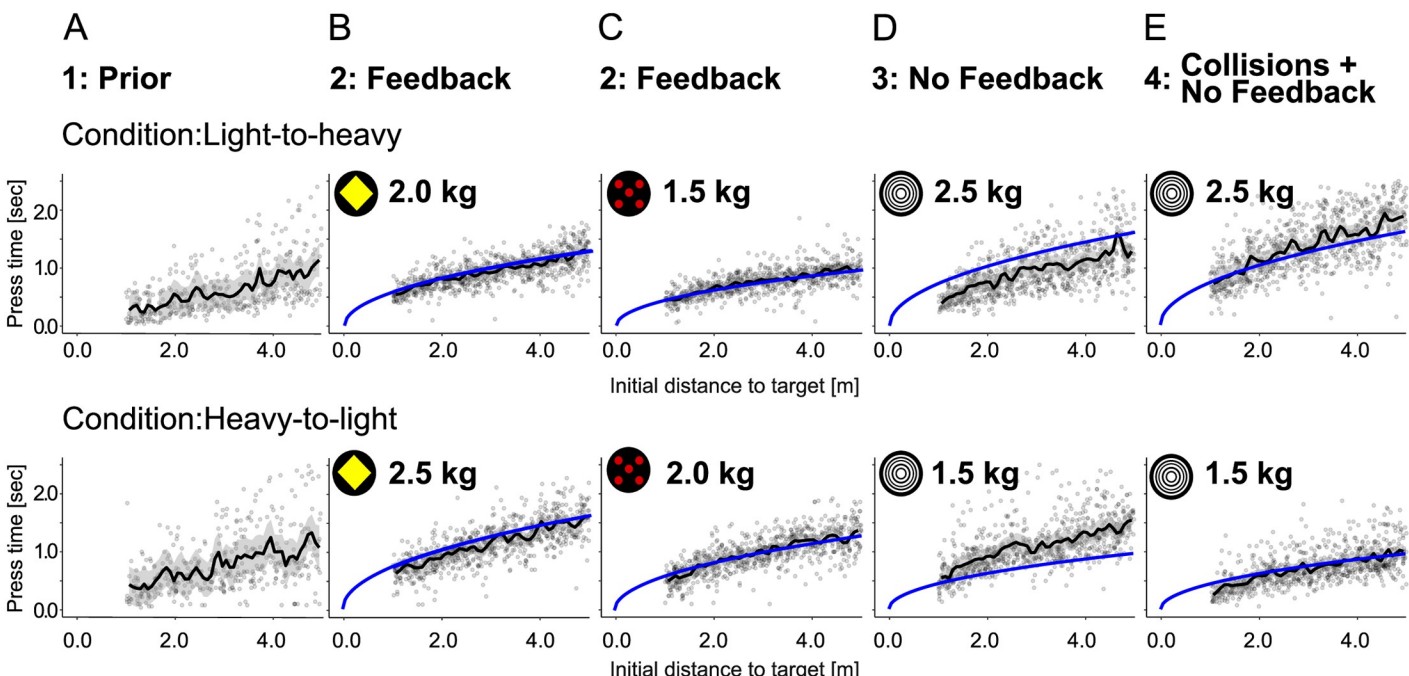

**Fig 2. Press-times as function of initial distance to target.** Press-times for all participants by condition and experimental phase are shown with data points in black and Newtonian relationship with perfect knowledge about the involved parameters in blue. The top row shows the data of subjects in the light-to-heavy condition and the bottom row shows the data of subjects in the heavy-to-light condition. **(A)** Press-times of participants in the first phase ("prior"), **(B)** second phase ("feedback") for the yellow puck, **(C)** second phase ("feedback") for the red puck, **(D)** third phase ("no feedback"), and **(E)** last phase ("collisions and no feedback") after having seen 24 collisions.

the press-time according to the square-root of the distance as specified in Eq 1. The correlation between press-times $t^{pre}$ and the square-root of the initial distance was 0.478 ($p < 0.001$). We expected the standard deviation of press-times to scale with the the mean of press-times in accordance with the Weber-Fechner scaling. This was confirmed by subdividing the range of distances into three intervals of the same size, i.e. $[1, 2.33]m$, $(2.33, 3.66]m$, and $(3.66, 5]m$ and computing the standard deviation of press-times within these three intervals resulting $2.97 \times 10^{-1}$ s, $4.19 \times 10^{-1}$ s, and $5.69 \times 10^{-1}$ s.

In phase two, participants adjusted their press-times based on observing the gliding of the pucks after button presses. Performance was evaluated by calculating the mean absolute distance of pucks to the target after sliding. The mean absolute error over the entire phase was $0.928m$ ($0.0177m$ SEM), see Fig 3. Accordingly, the correlation between $t^{pre}$ and the initial distance was 0.644 ($p < 0.001$) and with the square-root of distance 0.646 ($p < 0.001$). The performance improved between the first eight trials at the beginning of the phase (mean absolute error $1.76m$) and the last eight trials at the end of the phase (mean absolute error $0.89m$). The adjustment of pressing times was achieved on average after only a few trials, as revealed by a change-point analysis [29], which showed that after six trials the average endpoint error of the puck was stable (see S1 Appendix, "Change point detection"). Note that this includes four trials with one puck of the same mass and two trials of the second puck with a different mass.

Phase three involved shooting a new puck, which subjects had previously not interacted with, without visual feedback. Note that the puck was identical to the puck subjects later interacted with in phase four after seeing the collisions. This phase therefore allowed testing whether subjects used the non-linear scaling of the press-times depending on initial distance of the puck after having observed the pucks' motion in phase two. As expected, performance

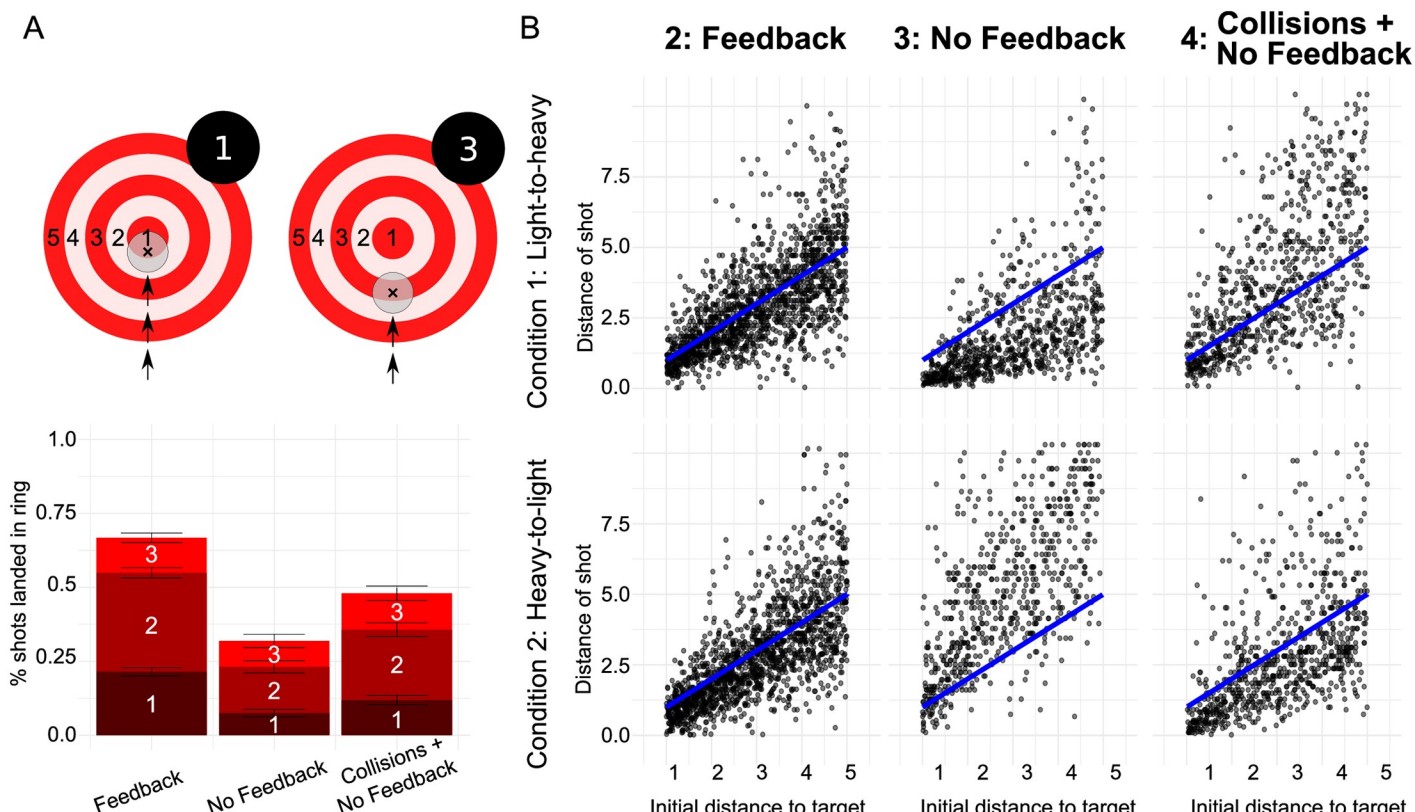

**Fig 3. Task performance and pucks' traveled distance for three phases of experiment. (A)** Participants' performance by experimental phase as quantified by pucks' average absolute error in final position. The number of the ring at which the center of the puck stopped was used for coding performance, e.g. 1 and 3 in the shown cases. **(B)** Aggregated final positions of pucks versus initial distance of pucks to target. Phases of the experiment are separated by columns and conditions are separated by rows. The line of equality representing final positions prescribed by the Newtonian model with perfect knowledge of all parameters is shown in blue.

was significantly lower with the new puck without obtaining visual feedback. Mean absolute error was $2.87m$ ($0.104m$ SEM), see Fig 3. The correlation between $t^{pre}$ and the initial distance was $0.599$ ($p < 0.001$) while the correlation between $t^{pre}$ and the square-root of the initial distance was $0.603$ ($p < 0.001$). Given that subjects had already obtained feedback about two pucks in phase two but did not obtain feedback in this phase, their press-time distribution could potentially be the mixture of the two press-time distributions of the two previous pucks, which were different in the conditions 'light-to-heavy' and 'heavy-to-light'. We compared the combined press-time distributions of phase two with the press-time distribution of phase three for each condition with the Kolmogorov-Smirnov test. Press-times in phase three reflected the behavior of both previous pucks combined for condition 'heavy-to-light' (Kolmogorov-Smirnov, D = 0.0538, p = 0.092, see S1 Appendix, "Kolmogorov tests—press-times in phase two & phase three") and approximately for condition 'light-to-heavy' (Kolmogorov-Smirnov, D = 0.156, $p < 0.001$, see S1 Appendix, "Kolmogorov tests—press-times in phase two & phase three").

At the beginning of phase four subjects watched a movie showing 24 collisions between the pucks from phase two, for which visual feedback of the gliding had been available, and the unknown puck from phase three. Thus, this condition allowed testing whether observation of the collisions was used to infer the mass ratios of pucks and to subsequently adjust the pressing times for that puck from phase three. Performance was significantly higher than in phase three

with a mean absolute error of $1.63m$ ($0.0440m$ SEM), although the puck was the same as in phase three and although subjects did not obtain visual feedback, see Fig 3. This effect was significant for both conditions as tested with Wilcoxon Signed Rank test for the absolute error (light-to-heavy: W = 339300, p = 0.018; heavy-to-light: W = 441330, p < 0.001). This shift towards longer and shorter press-times in the light-to-heavy and heavy-to-light condition respectively is depicted in S1 Appendix, "Press-time distributions". The shift was statistically significant by testing with a Wilcoxon Signed Rank test for shorter and longer press-times for both conditions respectively (light-to-heavy: W = 158580, p < 0.001; heavy-to-light: W = 490620, p < 0.001). For more detail of the error distributions across phases two to four see S1 Appendix, "Distance error distributions".

Taken together, these analyses suggest, that subjects adjusted their press-times both depending on the distance of the pucks to the target and depending on the pucks' masses used in the simulation. Furthermore, the analyses provide a very weak initial hint that subjects may have scaled their press-times with respect to mass and with a non-linear function of initial distance after having obtained visual feedback about the pucks' motion. Finally, observing collisions between pucks lead subjects to adjust their press-times even without obtaining visual feedback. In the following section we provide two computational generative models, one for the sliding task and one for the collision observation task to quantitatively analyze participants' press-times in terms of perceptual, physical, and behavioral quantities.

## Interaction model results

The above analyses give only a weak indication that our participants were able to adjust their press-times consistent with Newtonian physics and that they transferred the inferences about relative mass ratios from observing collisions to the press-times, and are limited in several ways. First, perceptual variables such as the initial distance of the puck to the target were uncertain for our subjects, which is not quantitatively entering the correlation analyses of press times with physical predictions under the assumption of perfect knowledge of all parameters. Secondly, our participants had to press a button to propel the puck. For longer press-times, subjects are known to demonstrate variability in pressing times, which scales linearly with its mean and which may vary considerably between subjects. Thirdly, while subjects pressed a button and observed the simulated motion of the pucks from a bird's eye view on a monitor, it would be desirable to be able to estimate subjects' belief about the masses of the different pucks implicit in their press-times. Therefore, we devised a hierarchical Bayesian model of the full visuomotor decision task to provide a computational account of our subject's behavior.

The Bayesian network model in Fig 4 expresses the relationship between variables on a subject-by-subject and trial-by-trial basis. While as experimenters we have access to the true initial distance $x$ used in the simulation of the puck and displayed on the monitor as well as the measured press-time $t^{pre}$ chosen by the subject on a particular trial $i$, subjects themselves do not know these values. Instead, each participant $j$ has some uncertain percept of the puck's distance $x_{i,j}^{per}$ and, potentially, some belief about the mass $m_{j,k}$ of the puck, which depends on its color and the phase of the experiment $k$. This structure of the graphical model from the experimenter's view leads to the following joint distribution $p(d, l)$ with observed data $d = \{x, t^{pre}\}$ and latent variables $l = \{x^{per}, \sigma^x, m, \sigma^t\}$, where trial, puck and participant subscripts were omitted for clarity:

$$p(d, l) = p(x) \; p(\sigma^x) \; p(x^{per}|x, \sigma^x) \; p(m) \; p(\sigma^t) \; p(t^{pre}|x^{per}, m, \sigma^t, \theta) \tag{2}$$

Here, $p(x)$ is known to the experimenter as the actual distribution of distances to target used in

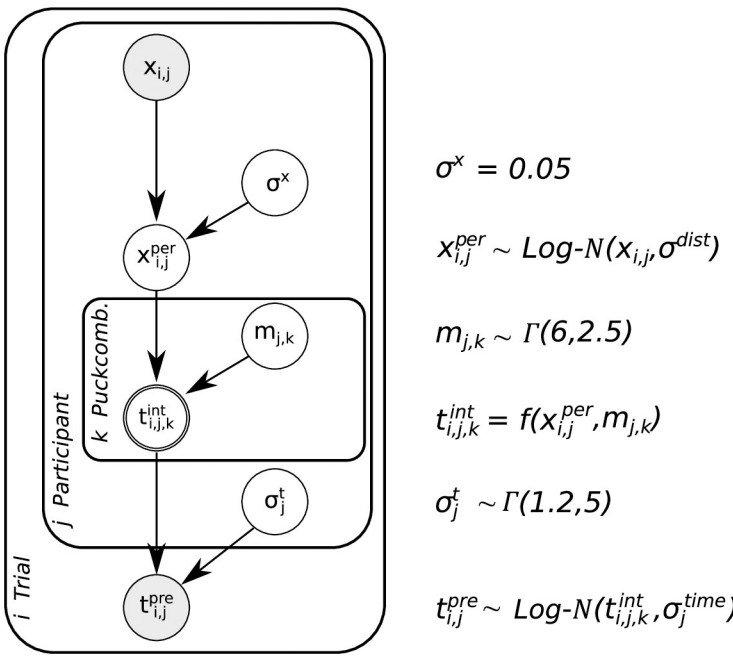

$\sigma^x = 0.05$

$x_{i,j}^{per} \sim Log\text{-}N(x_{i,j}, \sigma^{dist})$

$m_{j,k} \sim \Gamma(6, 2.5)$

$t_{i,j,k}^{int} = f(x_{i,j}^{per}, m_{j,k})$

$\sigma_j^t \sim \Gamma(1.2, 5)$

$t_{i,j}^{pre} \sim Log\text{-}N(t_{i,j,k}^{int}, \sigma_j^{time})$

**Fig 4. Hierarchical Bayesian network for the Newtonian interaction model.** The model expresses the generative process of observed press-times $t_{i,j}^{pre}$ across trials $i$, participants $j$, and pucks $k$ including Weber-Fechner scaling given perceptual uncertainties of distance $x_{i,j}$ and mass $m_{j,k}$ of the pucks and subjects' press-time variability. The parameter values refer to the prior probability distributions. See the text for details.

the simulations. By contrast, the distribution of perceived distances $p(x^{per}|x, \sigma^x)$ is the noisy perceptual measurement by our participants described as a log-normal distributed variable, ensuring that samples are strictly positive and including uncertainty scaling according to Weber-Fechner [30]. $p(\sigma_x)$ describes the prior distribution over possible values of this perceptual uncertainty. Participants' prior beliefs about the masses of the different pucks $p(m)$ are described by gamma distributions, which entail the constraint that masses have to be strictly positive. The log-normal distribution of actually measured press-times $p(t^{pre}|x^{per}, m, \sigma^t)$ depends on the noisy perception of the distance to target $x^{per}$, the belief about the mass of the object and the variability in acting, which is the press-time variability $\sigma^t$ with its gamma distribution $p(\sigma^t)$. We additionally summarize all constant factors, i.e the surface friction coefficient, the gravitational acceleration, the constant interaction force in the parameter $\theta$.

The potential functional relationship between the perceived distance of the puck to the target and the required press-time is expressed in the deterministic node representing $t^{int}$ in the Bayesian network. We consider two possible functional relationships between the press-time and the distance to be covered: subjects may use a linear relationship between press-time and initial distance as a simple heuristic approach:

$$H_1: \quad t^{int} \propto x^{per} \tag{3}$$

or may use the square-root relationship as prescribed by Newtonian physics according to Eq 1:

$$H_2: \quad t^{int} \propto \sqrt{x^{per}} \tag{4}$$

As experimenters, we only have access to the observed data $d$, i.e. the actual distances given the experimental setup and the measured press-times. We use Bayesian inference employing Markov-Chain Monte-Carlo to invert the generative model and infer the latent variables

describing subjects' internal beliefs given the observed data $d$:

$$p(l|d) = \frac{p(d,l)}{p(d)} = \frac{p(\sigma^x)\ p(x^{per}|x,\sigma^x)\ p(m)\ p(\sigma^t)\ p(t|x^{per},m,\sigma^t,\theta)}{p(t|x)} \tag{5}$$

However, modeling perception as inference may not be sufficient to describe our participants' behavior and their selection of actions. Given a posterior over mass and distance describing the perceptual belief of a subject on a particular trial, a specific press-time needs to be selected. In order to model this selection process we take action variability and potential cost functions into account. Cost functions govern which action, here the press-time, should be chosen given a posterior belief. Specifically, the cost function quantifies how the decision process penalizes errors on the task. This means that it is assumed that participants select an action that minimizes potential costs associated with missing the target. Loss functions, describing the rewards or costs for every action in the action space, can have any arbitrary form, nonetheless we chose a set of three standard loss functions and compare their predictions: 0-1, absolute and quadratic loss functions. These three canonical loss functions express subjects' implicit preferences for reaching a decision about press-times based on a putative perceptual posterior: the 0-1 loss corresponds to penalizing equally all deviations between the chosen value and the correct value, the absolute loss corresponds to penalizing deviations from the true value linearly, and the quadratic loss penalizes the deviations quadratically. It can be shown that these loss functions lead to different decisions for a continuous variable with a non-symmetric distribution [31]. Applying these three cost functions to a log-normal posterior results in the optimal decision being the MAP in case of the 0-1 loss function, the median for the absolute loss function, and the mean in case of the quadratic loss function. Thus, assuming that humans do have costs for missing the target and associated policies to minimize these costs, leads to three different model versions for each model class (see S1 Appendix, "Implementation of cost functions").

In order to evaluate participants' behavior computationally we first utilized subjects' data from phase two of the experiment to estimate their perceptual uncertainty and behavioral variability. We chose to start with analyzing phase two for two reasons: first, if participants are able to use visual feedback about the pucks' dynamics to adjust their press-times, predictions of the model with the correct physical relationships should capture the behavior better than the linear heuristics model. Secondly, inferred values for latent variables describing visual uncertainty in distance estimation and variability in press-times are less prone to be assigned additional uncertainty. Additional uncertainty arising in all other phases of the experiment due to the lack of visual feedback should be assigned to the uncertainty about the mass or the linear scaling rather than to the variability of press-times in general. Therefore, by evaluating data from phase two "feedback" first, values for the press-time variability and uncertainty in the perception of distances can be estimated for each participant.

First, we used the data of phase two "feedback" to investigate, which of the three loss functions best describes our participants' data. In order to choose the appropriate cost function explaining participants' actions most accurately, we computed the press-times predicted by the linear heuristics and the Newtonian model and applied the three cost functions to both models. This was achieved by using the inferred maximum a posteriori (MAP) values for the latent variables in both model classes, i.e. the mass $m$ in the Newtonian and a linear factor in the heuristic linear model class. This allowed calculating the residuals, i.e. the difference between subjects' actual press-times and the predicted press-times for all six combinations of two models and three cost functions. The residuals are shown as a function of the distance to the target in Fig 5. The strong correlation of residuals and distance to target indicates a systematic bias of

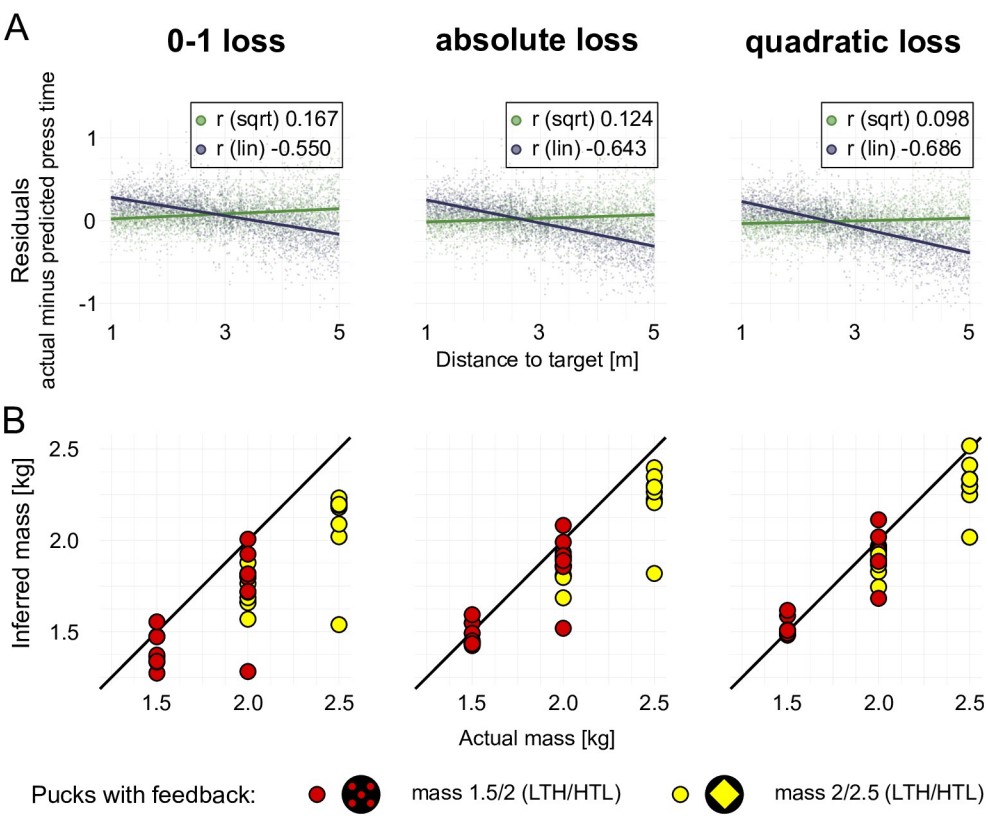

**Fig 5. Residuals of estimated press times and inferred masses in phase two for three cost functions. (A)** Residuals were calculated for each participant and each puck in phase two ("feedback") given the actual press-times and the best fits for the linear heuristics and the Newtonian model. Residuals for both models were calculated for all three cost functions. **(B)** MAP estimates of the masses used by individual subjects inferred according to the Newtonian model for the the three cost functions. Red and yellow pucks had different masses for subjects in the two conditions "heavy-to-light" and "light-to-heavy".

the linear heuristics model, whereas the weak correlation of the Newtonian model demonstrates its superiority in explaining the measured data. These relationships were tested with Spearman correlation tests for each model and cost function. The data show highly significant correlations for all models (p < 0.001 in all cases; 0-1 loss function: $\rho_{New} = 0.167$, $\rho_{lin} = -0.550$; abs. loss function: $\rho_{New} = 0.124$, $\rho_{lin} = -0.643$; quadratic loss function: $\rho_{New} = 0.0976$, $\rho_{lin} = -0.686$) and higher correlation in the linear model for each cost function (p < 0.001 in each case, with Bonferroni corrected $\alpha_{crit} = .017$).

Secondly, the posterior predictive distributions for press-times estimated from data in phase two (see S1 Appendix, "Posterior predictive checks for press-times") match the actual behavior of the participants more closely compared to the linear heuristics model. Kullback-Leibler divergence for each pair support this with divergence values at 0.0558 and 0.0851 for the Newtonian and linear model, respectively. Not only did the Newtonian model capture participants' press-times in phase two better than the linear heuristics model, but this also affected the inferred variabilities. While perceptual uncertainty only varied marginally (see Fig 6(A)), the posterior distributions of the press-time variability $\sigma_j^t$ show higher values for the linear model (see Fig 6(B)) compared to the Newtonian model. This was confirmed by calculating a repeated measure ANOVA on the posterior distributions of press-time variability for both models, showing that the difference was highly significant (F = 39.2, p < 0.001). This elevated

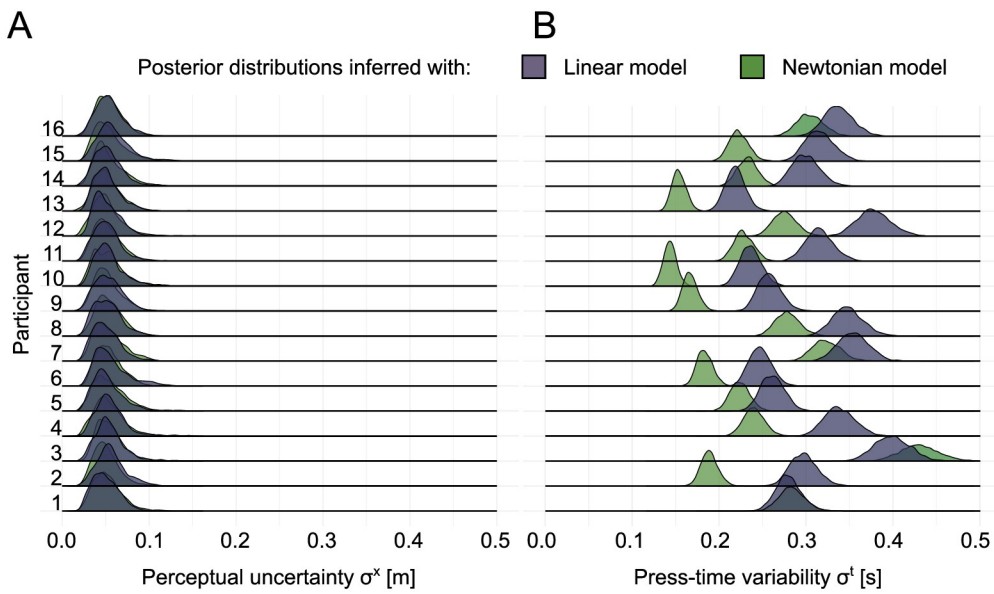

**Fig 6. Posterior estimates of perceptual uncertainty and press-time variability inferred with data from phase two "feedback". (A)** Inferred posterior distributions of perceptual uncertainty for the linear heuristics model and the Newtonian physics model. Dark green distributions display posterior distributions for the Newtonian model class, dark blue ones for the linear model class. A separation into cost functions is not included since the different cost functions did not lead to significant differences. **(B)** Inferred posteriors for individual press-time variability varied significantly between subjects between the two models. All but one participant show lower or equal values of variability regarding the press-time for the Newtonian model class.

level of uncertainty is necessary for the linear heuristics model to compensate for the diminished ability to capture the relationship of initial distances and participants' press-times. Therefore, in the following analysis we used the Newtonian model with quadratic cost, because it shows the lowest residual correlation, smallest divergence in posterior predictive distributions of press-times, and smallest press-time variability.

A consequence of selecting the quadratic cost function on the basis of the analyses of press-time residuals and posterior predictive distribution of press-times allows comparing the masses inferred on the basis of participants' behavior. Remarkably, posterior distributions inferred with data aggregated over participants only from phase two match actual masses implemented in the physical simulations better for the quadratic cost function (see Fig 5(B) and S1 Appendix, "Latent masses by cost function: aggregated data from phase 'feedback'"). In both conditions inferred beliefs about the masses are closer to the actual masses implemented in the simulations when presuming that participants use a quadratic loss function. This was confirmed by testing for the absolute differences between the posterior belief and the actual mass for each condition, puck and cost function. An ANOVA revealed highly significant differences (F = 486, p < 0.001) and post-hoc tests showed that the posterior belief when using the quadratic cost function is the closest fit for all pucks (p < 0.001 condition light-to-heavy, yellow diamond puck; p = 0.002 red dots puck; p < 0.001 condition heavy-to-light, yellow diamond puck; p < 0.001 red dots puck). This result also held at the individual participant levels as illustrated in Fig 5(B)). Thus, the quadratic cost function, which best described participants' press times, revealed that participants' mass beliefs were more accurate compared to assuming other cost functions.

Subsequently, we used the MAP values of the inferred press-time variabilities $\hat{\sigma}^t_{MAP}$ for each subject as fixed values for the analyses of data of all experimental phases. The same applied for

the MAP values of the inferred perceptual uncertainties $\hat{\sigma}^x_{MAP}$ which did not differ across subjects or models (see Fig 6(A)) and therefore were set to one fixed value for all subjects. Note that the mean was $0.05m$ in simulation space, which, given the current setup corresponded to approximately 4.7 pixels on the monitor. Using the hierarchical Bayesian interaction model, samples of the posterior predictive distributions of press-times and of the perceptual uncertainty are used to infer latent variables for both the linear and the Newtonian models. The posterior predictive distributions of press-times are shown in the S1 Appendix, "Posterior predictive checks for press-times in both models". Evidence was in favor of the Newtonian model compared to the heuristics model across all phases of the experiment with the exception of the *Prior* phase. The largest differences in prediction power appears in the *Feedback* phase with the Newtonian model being the considerably better choice to describe the actual press-times. This superiority of the Newtonian model over the linear heuristic one remains in the subsequent phases even without any visual feedback. This was again tested by running two-sample Kolmogorov-Smirnov tests for posterior predictive distributions of phase three of both models and the actual data, as well as calculating the Kullback-Leibler divergence for each pair, resulting in lower K-S statistic values for the Newtonian model (D = 0.0436, p = 0.00521) compared to the linear one (D = 0.0851, p < 0.001). KL divergence values are 0.0582 and 0.0599 for the Newtonian and linear model, respectively.

Finally, to confirm that the behavioral data of our subjects was best described by the Newtonian model with quadratic cost function we carried out model selection using the *product space method* [32]. In this approach, a mixture model combines both the linear and the Newtonian model to account for the data. An index variable indicates, which of the two models is selected at each iteration to explain the data. Given that both models have the same a priori probability to be chosen, the Bayes factor equates to the posterior odds of the index variable. Resulting Bayes factors are shown in Fig 7. Given the complete data set from all phases there is small support for the Newtonian model (Bayes factor $K$ of 2.33). When only considering data from the *Prior* phase there is weak support for the linear model ($K$ = 1.88). Instead, when

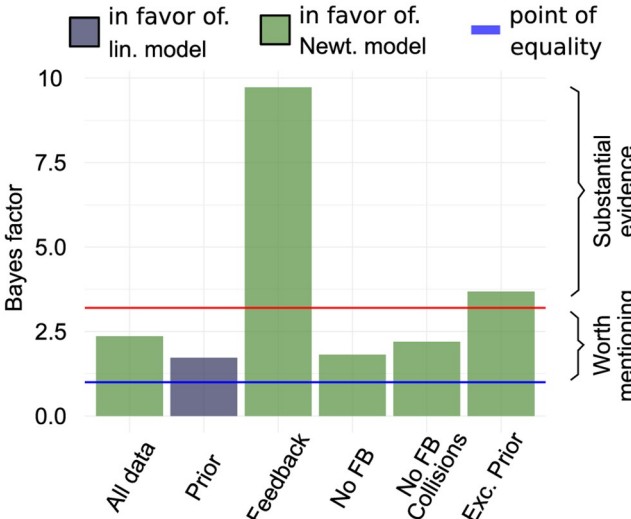

**Fig 7. Bayes factors calculated from posterior odds sampled using the product space method.** Bayes factors are displayed for different phases and combinations of phases. Blue line at 1 marks the point where neither model is stronger supported by evidence. Red line at 3.2 marks the transition from Bayes factors being only worth mentioning to substantial evidence in favor of one the models. Colors of bars indicate the model favored by the Bayes factors.

considering all phases but the first phase there is substantial support for the Newtonian model ($K$ = 3.71) and strong evidence for the square-root model in the feedback phase ($K$ = 9.71).

The hierarchical Bayesian interaction model also allows inferring the masses best describing our subjects' internal beliefs given the Newtonian model and the measured press-times. Not surprisingly, mean mass beliefs vary strongly across subjects in the *Prior* phase, where participants had to make decisions without any observations of the pucks, only relying on their prior beliefs about the potentially underlying dynamics and environmental conditions (see S1 Appendix "Latent masses: phase 'prior' and 'feedback'" for gray posterior distributions). Nevertheless, the variances of mass beliefs within the first phase were surprisingly small for individual subjects with a mean of 0.0023 kg, potentially indicating that each subject consistently used a belief about the mass of the puck. Inferred values for these prior mass beliefs are displayed in the S1 Appendix "Latent masses: phase prior and feedback" for each participant. When obtaining visual feedback in the *Feedback* phase of the experiment, subjects only needed on average six trials to adjust their press-times so that mass beliefs were stable thereafter. Implicit mass beliefs were quite accurate with the mean of inferred MAP values at 1.5218 and 1.8818 kg in the condition light-to-heavy (1.5 and 2.0 kg) and 1.9415 and 2.3068 kg in condition heavy-to-light (2.0 and 2.5 kg). Fig 8 shows the MAP estimates of the masses for both conditions and phases two to four for all subjects.

In phase three *No Feedback* participants faced an unknown puck without any visual feedback but with the acquired knowledge about the relationship of press-time and distance. Note however, that participants had learned two different mappings from distances to press-times in phase two, one for the red puck and one for the yellow puck. Thus, participants had to select press-times without knowing the mass of the unknown puck. As reported above, the press-time distributions in this phase of the experiment were close to the combined press-times that subjects had used for the two pucks in the previous phase two of the experiment. The corresponding MAP mass beliefs were accordingly approximately the average of the two previous pucks' masses with 1.87 and 2.19 *kg* and corresponding mass distributions differed significantly for the two conditions light-to-heavy and heavy-to-light (ANOVA: F = 1060, p < 0.001; see also S1 Appendix, "Latent masses: phase "no feedback" and "collision and no feedback""). But

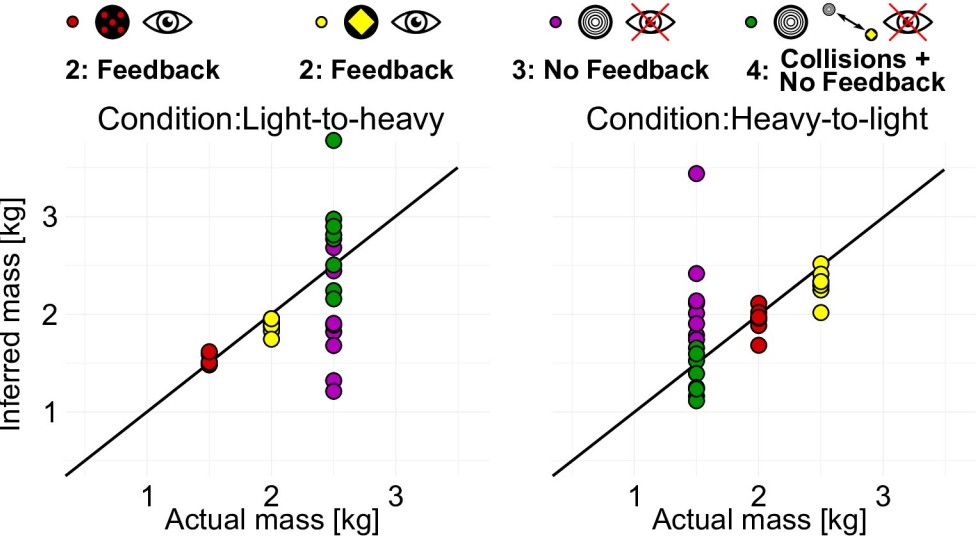

**Fig 8. MAP values of inferred latent mass in Newtonian model class with quadratic loss function for each participant and condition.**

after observing the 24 collisions in phase *Collisions + No Feedback* of the two known pucks with the unknown puck participants were able to adjust their press-times so that the estimated mass beliefs were significantly closer to the true values used in the simulations than in the previous phase. This was quantified by running a repeated measures ANOVA of the deviations from the actual mass ($F = 7.103$, $p = 0.0176$). Thus, the mass beliefs implicit in our participants' press-times reflected the inferred mass ratios and transferred from having observed the pucks' collisions to the subsequent visuomotor control task. Note that this implies that subjects must have interpreted the dynamics of the red and yellow pucks in the second phase as stemming from objects' masses, as otherwise a physically consistent transfer to a new puck would be very difficult to explain.

## Observation model result

Participants in our experiment were apparently able to make appropriate inferences in phases with feedback, altering their beliefs about unknown objects based on previous inferences and new observations, and to transfer this knowledge to an action-control task. But how were they able to make these adjustments after observing collisions and perform well with a continuous range of responses? Here, we want to look at another Bayesian model capturing the learning process through observations. To this end, we adapted a hierarchical Bayesian observation model similar to [9, 11], which describes how subjects could infer the relative mass ratios of two pucks from observing their elastic collisions. But here we used the mass beliefs inferred from phase two of the experiment with the interaction model as initial prior mass beliefs in the observation model for phase four of the experiment on-a-subject-by-subject basis. This allows comparing how subjects' uncertainty decreases on the basis of perceptual observations compared to visuomotor interaction.

The Bayesian network model for the observation task in Fig 9 expresses the relationship between variables on a subject-by-subject basis for observing 12 collisions for each of the two pucks. The model incorporates the generative physical relationship of velocities and masses in elastic collisions as shown in [9]. The grey nodes are known to the experimenter: the initial velocities $v_F$ and the mass $m_F$ of the known feedback puck and $v_{NF}$ of the unknown no-feedback puck, the resulting velocities $u_F$ and $u_{NF}$. Individual subjects' posterior mass beliefs at the end of phase two inferred with the interaction model, shown on the left panel of Fig 9, were used as prior mass beliefs of the yellow and red pucks in the observation model for each participant. Unknown parameters are depicted as white nodes and were inferred with MCMC. Subjects' uncertain beliefs about the pucks' velocities are incorporated for the initial velocities $v_F$ and $v_{NF}$ as well as for the resulting velocities $u_F$ and $u_{NF}$ after the elastic collision. To describe the perceptual uncertainty of velocities we used a log-normal distribution with $\sigma_{vel}$ fixed at 0.2 and its mode at the actual velocity (see Fig. 6 in [9] or section "Subject Performance" in [11] for comparison). Inferred posterior mass beliefs for the new puck are shown in the right panel. This structure leads to the following joint distribution $p(d, l)$ with observed data $d = \{v_F, v_{NF}, u_F, u_{NF}, m_F\}$ and latent variables $l = \{v_F^{per}, v_{NF}^{per}, u_F^{per}, u_{NF}^{per}, m_{NF}\}$, where actual and perceived velocities are summarized for both pucks using an index $i$ to $v_i$ and $u_i$ for abbreviation purposes:

$$p(d, l) = p(v_i)\ p(u_i)\ p(m_F)\ p(v_i^{per}|v_i, \sigma_{vel})\ p(m_{NF})\ p(u_i^{per}|u_i, v_i^{per}, m_F, m_{NF}, \sigma_{vel}) \qquad (6)$$

The observation model allows inferring participant's mass beliefs for the puck, which they had first interacted with in phase three of the experiment. Importantly, the two Bayesian models allow inferring the uncertainty in participants' mass beliefs after only six and after 24 trials,

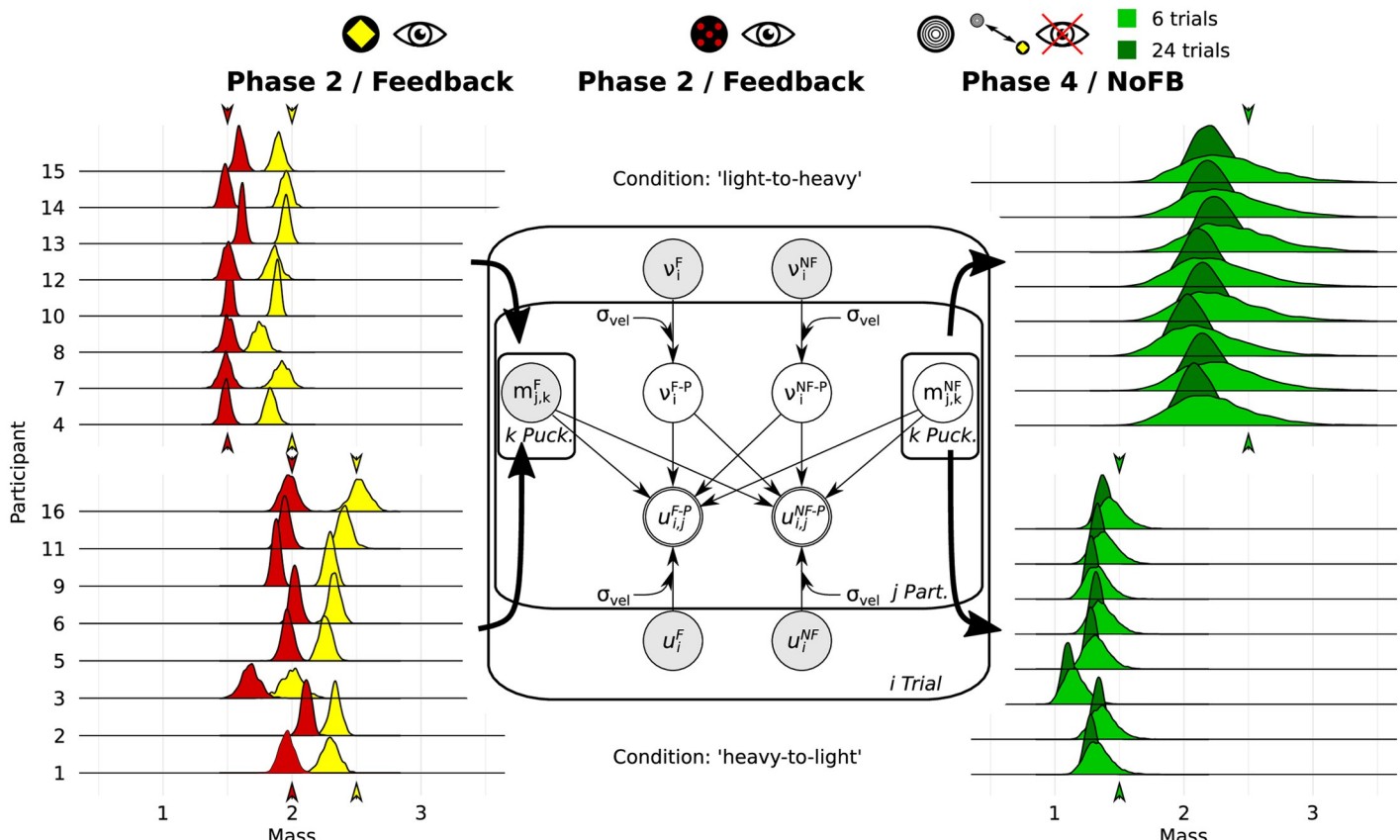

**Fig 9. Bayesian model for learning through observing collisions with prior and posterior mass beliefs.** The left panel shows inferred posterior mass beliefs for the pucks from feedback phase 2 for each participant. All 100 trials were used to infer the mass beliefs. These posteriors were used as priors for the inference from observations. The graphical model for learning by observing collision is shown in the middle panel. Uncertainty about the pucks' velocities is introduced for the initial velocities $v_F$ and $v_{NF}$ as well as for the resulting velocities $u_F$ and $u_{NF}$ after the elastic collision. Utilizing the physical relationship of velocities and masses in an elastic collision enables inferring beliefs about the unknown puck based on previous mass beliefs of pucks in phase 2. Resulting posterior mass beliefs are shown in the right panel for inferences based on 6 and 24 observations of collisions.

both for the interaction phase two and the observation of the collision movies, see S1 Appendix, "Learning progress of mass beliefs during interaction and observation". These results quantify, how uncertainty in mass beliefs decreased over trials and the difference in uncertainty reduction due to interactions versus observations. More specifically, as expected, subjects' variance in inferred posterior mass beliefs for each puck decreased with the progression of trials when using the interaction model with data from phase 2 (Friedman chi-squared = 62.06, p-value < 0.001 & Conover's PostHoc p < 0.001 for all comparisons) and, as well, when using the observation model with mass beliefs from phase 2 with the highest precision after 100 trials (Wilcoxon signed rank test, V = 136, p < 0.001). Additionally, the variance in resulting inferences about the mass in the observation model is significantly higher than the variance of the mass beliefs used as input, as we compared variances on subject basis for columns three, four and five (Kruskal-Wallis chi-squared = 37.43, p-value < 0.001 & Dunn PostHoc for grey compared to red and green, each p < 0.001, see S13 Fig). Thus, the larger variance in participant's mass estimates after observing the pucks' collisions compared to interacting with them, see e.g. Fig 8, stems from the fact that subjects needed to use the uncertain mass beliefs of the red and yellow pucks when observing the collisions and had additional uncertainty stemming from inferring pucks' velocities. Furthermore, the predictions of the idealized

observation model deviate quantitatively from mass beliefs inferred using the interaction model for two reasons: First, participants would need to remember their belief about the mass of both feedback pucks perfectly while performing in phase 3 and 4. However, these beliefs may suffer from memory effects and thus potentially introduce biases and additional variability. Second, initial and uninformed guesses in phase 3 before seeing any collisions may generate biases, too, that potentially could lead to recency effects (see e.g. participant 7 & 8 in S10 Fig).

## Discussion

Although people are able to interact with the physical world successfully in every-day activities, classic research has contended that human physical reasoning is fundamentally flawed [1–4]. Recent studies instead have shown that biased human behavior in a range of perceptual judgement tasks involving physical scenarios can be well described when taking prior beliefs and perceptual uncertainties into account [9–12]. The reason is that, inferences in general need to integrate uncertain and ambiguous sensory data and partial information about object properties with prior beliefs [5–8]. Much less is known about how intuitive physical reasoning guides actions. Here, we used a perceptual inference task involving reasoning about relative masses of objects from the intuitive physics literature and integrated it with a visuomotor task. Subjects had to propel a simulated puck into a target area with a button press whose duration was proportional to the puck's release velocity. The goal was to investigate how people utilize relative masses inferred from watching object collisions to guide subsequent actions.

Specifically, we devised an experiment consisting of four phases, which differed in the available sensory feedback and prior knowledge about objects' masses available to participants. The physical relationship underlying the task requires subjects to press a button for a duration that is proportional to the mass of the puck and proportional to the square-root of the initial distance. This allowed examining peoples' prior assumptions about the underlying dynamics of pucks' gliding, their ability to adjust to the pucks' initial distances to the target and to the varying masses of pucks, and the transfer of knowledge about relevant properties gained by observing collisions between pucks. A hierarchical Bayesian generative model of the control task and one of the collision observation task accommodating individual differences between subjects and trial by trial variability allowed analyzing subjects' press-times quantitatively. Importantly, we also tested which of three cost functions best describe our subjects' choices of press-times.

In the prior phase without visual feedback, subjects adjusted their press-times with the initial distance of the puck to the target. Not surprisingly, because subjects did not obtain any feedback about their actions and therefore the degree of friction, the magnitude of the applied force, and the scale of the visual scene, could only hit the target by chance. Nevertheless, model selection slightly favored the linear heuristics model compared to the square-root model, i.e. subjects approximately scaled the press-times linearly with the initial distance to target. Thus, subjects came to the experiment with the prior belief that longer press-times would result in longer sliding distances but did not scale their press-times according to the square-root of the initial distance of the pucks as prescribed by Newtonian physics. As subjects did not sense the weight of the pucks and did not obtain any visual feedback about the pucks' motion, the observed behavior in this phase of the experiment may be dominated by the uncertainty about the underlying mapping between the duration of button presses and the pucks' release velocities, the effects of friction, and the visual scale of the simulation. Remarkably, while no feedback was available, each participants' scaling of press-times was consistent as indicated by individuals' variance in posterior mass estimates being of the same order of magnitude as in feedback trials, see S1 Appendix, "Latent masses: phase 'prior' and 'feedback'".

When visual feedback about the pucks' motion during the feedback phase was available, subjects needed on average only six trials to reach stable performance. This is particularly remarkable, because it corresponds to adjusting the press-times to a single puck's mass over the four initial trials and then adjusting the press-times within only two subsequent trials to a new puck with a different mass. Thus, the observation of the pucks' dynamics over six trials was sufficient to adjust the press-times with the square-root of initial distance, but differently for the two pucks, see Fig 2. Note that in phase two, subjects only had a contextual color cue distinguishing the two pucks. Therefore, subjects needed to learn two different functions relating the pucks' initial distances to the required press-times, one for each puck, without any explicit reference to mass. Data from this phase of the experiment were utilized to infer parameters describing individual subjects' perceptual uncertainty and motor variability. Perceptual variability was consistent across subjects and varied only marginally so that a constant value of $\sigma^x = 0.05m$ was used across subjects and models for all other phases of the experiment. Remarkably, this corresponds to a distance of 4.7 pixels in the vertical direction on the display monitor with a resolution of 1080 pixels. By contrast, the variability of press-times $\sigma^t$ varied substantially across subjects with almost all subjects lying between $0.15s$ and $0.33s$, so that individuals' parameters were used in all subsequent models.

Given that the variability of peoples' press-times scales with the mean of the duration, longer press-times can lead to larger deviations from the targeted press-time. This can result in larger errors by overshooting the target. To reduce possible overshoots, participants may implicitly aim at a shorter distance, which can be quantified through a cost function incorporating the relative desirability of the pucks' final distance to the target. Therefore, we tested which of three commonly used cost functions best described subjects' press-times: the 0-1 cost function, the quadratic cost function, and the absolute value cost function. Model selection using the product space method showed that the press-times were best explained by the Newtonian physics model when taking into account perceptual uncertainty, motor variability and the quadratic cost function. Similarly, this was confirmed through posterior predictive checks of press-times and the analysis of the correlation of the residuals between predicted and observed press-times with the initial distance to target.

Thus, participants adjusted the press-times with the square-root of the initial distance to the target and used the contextual color cue of the pucks to adjust the press-times. Subjects only had the contextual cue of different colors between the two pucks but adjusted the press-times in such a way that this was interpretable in terms of the two different masses used in the puck's simulations. Therefore, just on the basis of these adjustments alone, one might argue that subjects may have adjusted their press-times based on the available visual feedback about the pucks' motion without any recurrence to a the concept of physical mass. That this is unlikely, is due to the following two phases of the experiment.

Previous research has demonstrated, that people can infer the mass ratios of objects from observing their collisions [9, 11–13]. Here, subjects were asked to propel one particular puck before and after seeing 24 collision between this puck and the two pucks for which they had previously obtained visual feedback. Note that the two pucks in phase two were only distinguished by a color cue and that subjects might have only learned two different mapping from initial distances to press-times, as no explicit cues about mass were available. But subjects readily utilized the inferred mass ratios to adjust their press-times to reach the target more accurately in phase four of the experiment. That the different dynamics were to attribute to different masses and that relative masses from observing the collisions could be transferred to press-times entirely relied on subjects intuitive physical reasoning. This is strong evidence that participants in our experiment interpreted the dynamics of the red and yellow pucks from phase two to be caused by their respective masses. Model selection provided evidence, that

subjects continued to use the square-root relationship of initial distance and scaled their press-times consistent with Newtonian physics to successfully propel the puck to the target.

Different from tasks requiring a forced choice response [1–4, 9, 11–13], participants in the current experiments provided a continuous action by pressing a button for variable durations. Therefore, it is not sufficient to model our participants' actions as in an inference task, e.g. by assuming that subjects choose a press-time on the basis of the mass belief with highest probability, i.e. the MAP. Instead, modeling continuous actions requires a cost function, which additionally incorporates people's variability in press-times. This is evident when comparing the press-times according to the different models considered here, see S1 Appendix, "Deviations from fully-observed Newtonian physics and model predictions". Remarkably, posterior means of masses best explaining our participants' press-times were closer to the true masses used in the pucks' simulations for the quadratic cost function compared to the other cost functions. Thus, the current study establishes that people's deviations from the predictions of Newtonian physics are not only attributable to prior beliefs and perceptual uncertainties but also to implicit cost functions, which quantify internal costs for errors due to participants' action variability.

Taken together, the present study is in accordance with previous studies on intuitive physics within the noisy Newton framework [14]. The systematic deviations in our subjects' press-times from the those prescribed by Newtonian physics under full knowledge of all parameters were explained quantitatively as stemming from perceptual uncertainties interacting with prior beliefs according to probabilistic reasoning. Previous studies had also shown, that people are able to infer relative masses of objects from their collisions [9, 11, 12]. The present study additionally shows, that subjects can utilize such inferences and transfer them to a subsequent visuomotor task. This establishes a connection between reasoning in intuitive physics [9–12] and visuomotor tasks [21, 23, 25, 27]. Crucial in the quantitative description of participants' behavior was the inclusion of a cost function. Commonly, cost functions in visuomotor behavior are employed to account for explicit external rewards imposed by the experimental design, e.g. through monetary rewards [21, 33] or account for costs associated with the biomechanics or accuracy of movements [23, 24]. The present model used a cost function to account for the costs and benefits implicit in our participants visuomotor behavior and may encompass external and internal cost related to different task components, perceptual, cognitive, biomechanical costs and preferences. Inferring such costs and benefits has been shown to be crucial for the understanding of visuomotor behavior [34–36].

The results of the present study furthermore support the notion of structured internal causal models comprising physical object representations and their dynamics. Although our participants never sensed the weight of pucks, they readily transferred their visual experiences by interpreting them in terms of the physical quantity of mass. A recent study [37] found support at the implementational level for representations of mass in parietal and frontal brain regions that generalized across variations in scenario, material, and friction. While our results do not provide direct evidence for the notion of internal simulations of a physics engine [38], they also do not contradict them. While it could be argued that structured recognition models may be sufficient for the inference of object properties such as mass, in our experiment subjects had to act upon such inferences, which strongly suggest the availability of representations of mass.

Finally, the present study also shows the importance of using structured probabilistic generative models that contain interpretable variables when attempting to quantitatively reverse engineer human cognition [39]. Previous research has demonstrated pervasive and systematic deviations of human reasoning from probabilistic accounts [40]. Similarly, systematic deviations in physical reasoning [1–4] have been interpreted as failures of physical reasoning. It is

only more recently, that a number of these deviations have been explained through computational models [9–12, 38] involving structured generative models relating observed and latent variables probabilistically. These models involve the explicit modeling of prior beliefs and perceptual uncertainties [5, 6] as well as uncertainties in visuomotor behavior [21–23], which have been modeled successfully in a probabilistic framework. As such, the present study is in line with efforts of understanding perception and action under uncertainty through computational models, which use structured probabilistic generative models and external as well as internal costs [8].

## Supporting information

**S1 Appendix. Puck motion.**
(PDF)

**S1 Fig. Distance error distributions.** Final discrepancy between target and puck pooled for all participants. Pucks being shot too short are shown with negative values, pucks with a positive deviation were shot too far. Columns showing the the data for both conditions and rows divide into puck and phase combinations. The first two rows (in gold and red) showing the error distributions for both pucks with feedback in phase 2. The error distribution for the unknown puck in phase 3 before seeing the collisions is shown in the second last row (in purple) with greater deviation, with a clear bias and bigger spread. In the last row the error distributions are depicted for the unknown puck after having seen the collisions with the previous learned pucks, showing a reduced bias.
(TIF)

**S2 Fig. Press-time distributions.** Pooled press-time distributions for all participants. Columns showing the the data for both conditions and rows divide into puck and phase combinations. First two rows showing the press-times for the pucks with feedback. Press-time distributions in phase 3 without feedback are shown in row three in blue. Without further information participants' behavior in phase 3 is strongly influenced by the previous phase and its press-time distribution: press-time distributions for the unknown puck in phase 3 reflect roughly the combined distributions of press-times of the previous pucks in phase 2 (Kolmogorov $D = 0.0538$; $p = 0.092$ for heavy-to-light, $D = 0.156$; $p = 9.8 \times 10^{-12}$ for light-to-heavy).
(TIF)

**S3 Fig. Kolmogorov tests—Press-times in phase 2 & phase 3.** In the light-to-heavy condition both distributions of press times when seeing pucks and without feedback in phase 3 differ significantly. However, considering the asymmetry within the task response—press-times and potential masses are only constrained single-sided towards lower values with a minimum at zero—this difference in press-time distributions is surprisingly small. **(B)** In the heavy-to-light condition there was no significant difference between the distribution of press-times of both combined feedback pucks and the unknown puck before observing the collisions as revealed by the Kolmogorov-Smirnov test. This suggests that participants adhere to their previous adjusted strategies when facing decisions in great uncertainty.
(TIF)

**S4 Fig. Implementation of cost functions.** Derivation of the three cost function models based on the expressions for the measures of the central tendency of the log-normal distribution with its mode $exp(\mu - \sigma^2)$, median $exp(\mu)$ and mean $exp\left(\mu + \frac{\sigma^2}{2}\right)$. Setting the intended press-time to one of these measures for the press-time distribution is equivalent with choosing the 0-1, absolute or quadratic loss function. Transformation with the intended press-time $t^{int}$ leads

to expressions in S4 Fig.
(TIF)

**S5 Fig. Posterior predictive checks of cost functions in phase 2.** Posterior predictive distributions for both model classes and all cost functions with data from phase 2 with feedback. Posterior predictive distributions of press-times given data from feedback trials. Fifty distributions were drawn from each model after being fitted to the data. Dark green distributions arise from models of the Newtonian model class, dark blue ones from the linear model class. Separation into rows is made on basis of the implemented cost function. For each cost function the Newtonian model predicts values that match the actual data shown as red curve obviously better than the model from the linear model class.
(TIF)

**S6 Fig. Posterior predictive checks for press-times in both models.** Posterior press-time predictions for both, the linear and the Newtonian model with quadratic cost function, and separately for every phase. Actual data is shown as red line. Model predictions in dark green (50 iterations) of the fitted Newtonian model match the data closely and surpass the fitted linear model in dark blue for the complete data set and in almost every phase individually.
(TIF)

**S7 Fig. Latent masses by cost function: Aggregated data from phase 2.** Inferred latent mass beliefs with aggregated data from phase 'feedback' for each cost function. Posterior distributions for mass belief aggregated over all participants for each cost function. Colored, vertical lines indicate actual mass of pucks. In comparison the quadratic loss function leads to posterior distributions that fit closest to the actual masses in the experiment.
(TIF)

**S8 Fig. Change point detection.** Average absolute error as function of trials and posterior of mean average error derived using the change point detection model. **(A)** Average absolute error over participants as function of trial number. **(B)** Posterior over change point $\tau$. Red dotted line marks trial six. **(C)** Posterior of mean error before and after change point.
(TIF)

**S9 Fig. Latent masses: Phase 'prior' and 'feedback'.** Inferred latent mass in Newtonian model class with quadratic loss function for each participant and with data from *Prior* and *Feedback* phase. Posterior mass distributions for each participant in *Prior* and *Feedback* phase. Gray distributions show the inferred mass distribution for an unknown puck before participants have encountered the task dynamics. Resulting mass distributions for both pucks in feedback trials in red (light puck) and yellow (heavy puck). Dotted lines indicate actually implemented mass for each of the feedback pucks.
(TIF)

**S10 Fig. Latent masses: Phase 'no feedback' and 'collision and no feedback'.** Inferred latent mass in Newtonian model class with quadratic loss function for each participant with data from *Prior* and both *No Feedback* phases. Posterior mass distributions for each participant in *Prior* and *Feedback* phase. Gray distributions show again the inferred mass distribution for an unknown puck before participants have encountered the task dynamics. Distributions in violet and green are the posterior mass distributions of the unknown puck without feedback before and after the participants saw collision with known pucks. Dotted line marks the actual mass of the unknown puck.
(TIF)

**S11 Fig. Deviations from fully-observed Newtonian physics and model predictions (light to heavy).** Posterior predictive for press times, actual press times and ideal responses for phases two to four and condition light-to-heavy. Black distributions show the actual data, red and blue ones display samples from posterior predictive distributions of both, the linear and Newtonian model, and green ones show the correct responses given perfect knowledge about the underlying physics and all parameters. Visualizing the enhanced suitability of this noisy Newtonian model framework compared to Newtonian models excluding prior preferences and uncertainties in describing human behavior.
(TIF)

**S12 Fig. Deviations from fully-observed Newtonian physics and model predictions (heavy to light).** Posterior predictive for press times, actual press times and ideal responses for phases two to four and condition heavy-to-light. Black distributions show the actual data, red and blue ones display samples from posterior predictive distributions of both, the linear and Newtonian model, and green ones show the correct responses given perfect knowledge about the underlying physics and all parameters. Visualizing the enhanced suitability of this noisy Newtonian model framework compared to Newtonian models excluding prior preferences and uncertainties in describing human behavior.
(TIF)

**S13 Fig. Learning progress of mass beliefs during interaction and observation.** Barplot of averaged variance for both models and a given number of observations. First three columns show the average variance in posterior mass beliefs for inferences with 6, 24 and 100 trials per puck and participant. Two last columns show the average variance of mass beliefs of the unknown puck resulting from inference using the collision model for 6 and 24 trials, while using the posterior mass belief of the known pucks from the interaction model with 100 trials each.
(TIF)

## Acknowledgments

We acknowledge support by the German Research Foundation and the Open Access Publishing Fund of Technische Universität Darmstadt.

## Author Contributions

**Conceptualization:** Nils Neupärtl, Constantin A. Rothkopf.

**Formal analysis:** Nils Neupärtl, Constantin A. Rothkopf.

**Investigation:** Nils Neupärtl, Fabian Tatai, Constantin A. Rothkopf.

**Methodology:** Nils Neupärtl, Fabian Tatai, Constantin A. Rothkopf.

**Visualization:** Nils Neupärtl, Constantin A. Rothkopf.

**Writing – original draft:** Nils Neupärtl, Constantin A. Rothkopf.

**Writing – review & editing:** Nils Neupärtl, Constantin A. Rothkopf.

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
