## [Decision Letter · Decision Letter 0]

15 May 2020

Dear Dr. Rothkopf,

Thank you very much for submitting your manuscript "Intuitive physical reasoning about objects' masses transfers to a visuomotor decision task consistent with Newtonian physics" for consideration at PLOS Computational Biology.

As with all papers reviewed by the journal, your manuscript was reviewed by members of the editorial board and by several independent reviewers. In light of the reviews (below this email), we would like to invite the resubmission of a significantly-revised version that takes into account the reviewers' comments.

----

While all the reviewers praise the rigour and quality of the paper, reviewer two is sceptical of the innovativeness. One suggestion by the reviewer is to ask for a Bayesian model that takes into account the subject inference process of the mass, as opposed to just model the inference of the distance and the action itself.

Currently, each subject infers the mass of the virtual pucks independently from the model. Incorporating the inference of the mass within the current model may be difficult, but adding a separate section with a separate model for the inference of the mass seems less onerous, and would still add novelty.

The editorial team would like to emphasise that this point should be addressed before we would consider the paper for publication.

----

We cannot make any decision about publication until we have seen the revised manuscript and your response to the reviewers' comments. Your revised manuscript is also likely to be sent to reviewers for further evaluation.

Sincerely,

Ulrik R. Beierholm

Associate Editor

PLOS Computational Biology

Wolfgang Einhäuser

Deputy Editor

PLOS Computational Biology

Reviewer's Responses to Questions

**Comments to the Authors:**

Reviewer #1: In this manuscript, the authors test whether physical knowledge transfers from visual inference to a motor planning task. They asked participants to hold down a button to launch a puck at a target, where the length of time the button was held was proportional to the velocity divided by the object mass. Participants progressed through four phases, from no feedback, to feedback with two objects of different masses, to no feedback with a new object, followed by showing that new object colliding with the old objects, then a final phase with no feedback. Crucially, participants adjusted their actions between the third and fourth phases to relatively accurately calibrate their push times for the new mass, even though they were only able to indirectly infer the mass through the collision observations. Thus, people appear to be able to transfer mass knowledge between visual and motor domains using relatively calibrated physics.

This paper demonstrates that people transfer physical knowledge across domains, a fact that has not been directly tested but is a prediction of recent theories that people have an Intuitive Physics Engine for generalizable physical reasoning. The experiment is relatively straightforward yet elegantly tests the main point of the paper, and the authors provide a thorough set of analyses and models to test their theories (to the point where multiple times I would think of supplementary analyses, then find them elsewhere in the manuscript). The only major comment I have is that while the authors do a good job showing mass transfer in this experiment, a secondary conclusion – that participants’ responses were “Newtonian” – is less well supported by the data. However, overall I believe this is a strong paper.

Signed,

Kevin Smith

-----

The secondary conclusion of this paper – that participants behaved in a “Newtonian” way – appears to be primarily motivated by model comparisons that test a model that includes a square-root relationship between press time and distance (Newtonian) against a model that assumes this relationship is linear. However, there is only clear evidence that the relationship is a square-root in the experimental phase with feedback – after that phase there is weak evidence in favor of the Newtonian model, whereas in the first phase there is equivalently weak evidence for the *linear* model (Fig. 7). The authors address this in the discussion as possibly stemming from unclear relationships between button pressing and force (pg. 16-17) which then gets corrected rapidly once feedback is available. I find myself agreeing with this interpretation and see it as the most parsimonious explanation. But a less charitable reader might suggest that the relationship between press time and distance is not based on physics at all, but rather is a functional mapping learned rapidly in phase 2; the physical reasoning about mass might then be the only physical part that then gets bolted onto the functional mapping in a sort-of Newtonian way.

I can think of three ways of adapting the text that might ground these claims better: (1) not referring to the square-root model as “Newtonian” and making clear that the Newtonian nature of physics is in the mass inference and transfer; (2) hedging with the Newtonian claims a bit more (e.g., the transfer is “consistent with Newtonian physics” as suggested by the title, but it’s not necessarily true that “participants transferred… in accordance with Newtonian physics” as suggested by the abstract); or (3) putting a section in the discussion with the least charitable, most non-Newtonian explanation that the authors can think of as an alternate model not tested here.

Alternately, it might be possible to show more evidence for mass transfer using Newtonian physics using an extension to the “inferred mass” analysis displayed in Fig. 8. It’s particularly compelling that individuals’ inferred mass is on average mostly calibrated to the actual mass. But it’s not clear that this requires extrapolation with Newtonian mechanics – would it be possible to extrapolate the constant factor in the linear model to what it “should” be if the mass ratios were extracted appropriately from the pre-phase 4 observations, and compare this to the linear model fits for individuals? If participants’ behavior deviated strongly from this value, then it suggests that people are extrapolating their action choices using Newtonian mechanics and not a linear heuristic.

More minor comments include:

• In the author summary, beginning of the introduction, and final paragraph of the discussion, the authors make the point that systematic failures of physical reasoning might be “the consequence of perceptual uncertainties and partial knowledge of object properties.” This is not quite true – many of the classical errors in reasoning cannot easily be explained by fully accurate physics (in fact, this failure to model a classic error was what prompted Smith, Battaglia, Vul, 2018). Instead, the findings tend to suggest that these errors might rely on distinct systems of reasoning to the more calibrated physics engine used for perceptual-motor tasks. It’s in these types of perceptual-motor tasks that errors appear to arise from uncertainty and partial knowledge.

• The authors present the number of trials used in Phase 1 and Phase 2 (pg. 6), but I was unable to find the corresponding information for Phases 3 and 4

• The error scale in Fig. 2B might mislead readers by making more erroneous points lighter and therefore less salient. Since this information is already present in the graph (as distance from the blue line), it might be better to use a constant alpha here.

• On pg. 8, the authors report correlations between distance and press times or square-root press times – what are readers to make of the fact that they are nearly identical in all four phases? It strikes me that many of the stimuli sit in a range where the quadratic function looks nearly linear (Fig. 3), and therefore my take-away is that people are modulating their press times with distance, but it’s not particularly diagnostic here of the exact functional relationship. This might also affect the statement on pg. 9, “analyses provided initial evidence, that subjects scaled their press-times… with a square-root function of initial distance”

• In Fig. 4 (pg. 10), values for model parameters are displayed, some of which are later fit and found to have little individual variability (sigma_x), or fit at an individual level (sigma_j^t). However I could not tell how the m_j,k parameter is fit, and sigma_j^t values don’t match up with those shown in Fig. 6. How were these values chosen? Are they meant to map on to the actual values used in the model?

• On pg. 11 for Eqs. 3 & 4, it could be useful to reiterate that these relationships only hold for a constant puck mass

• The sentence starting on pg. 12, line 396 might be more grammatical to start “The data show highly significant…”

• The sentence on pg. 13, line 412 should start “[An] ANOVA…”

• The sentence on pg. 14, line 452 should either be connected with the last sentence with a comma, or start “This results in lower…”

• In Fig. 11, which line represents the feedback vs. no feedback conditions?

• The data does not appear to be available on the authors’ lab website, but should be posted before publication

Reviewer #2: This paper examines the link between physical reasoning and visuomotor control. They perform an experiment in which participants press a button to launch pucks of varying mass at a target location with and without visual feedback. The experimental results reveal that participants quickly learn to appropriately control the puck, and that they can infer the mass of a new puck by just watching videos of it colliding with other pucks, and that they can use this knowledge of mass to more accurately control it. Through a Bayesian analysis, it is demonstrated that participants accurately estimate the mass and that they control the pucks according to Newtonian laws (i.e., by pressing the button for a duration that is a function of the square root of the distance it needs to travel) while also optimizing for internal costs/preferences.

Overall, I found this an enjoyable paper to read and believe it to be a useful contribution to the literature on intuitive physical reasoning, especially in terms of confirming a link between inference of physical properties and control. I found the paper relevant in that it supports the hypothesis that people ought to use their knowledge of physical properties to guide behavior. Moreover, it is extremely rigorous and thorough in demonstrating its results.

However, I thought there were three main shortcomings:

1. I do not feel that the paper contributes substantially new knowledge beyond that which already exists in the literature. There have been a number of other papers at this point showing that people can infer mass after observing physical events like collisions (e.g. Sanborn et al. 2013; Ullman et al. 2018), that they can make physical predictions based on knowledge of mass (Battaglia, Hamrick, & Tenenbaum, 2013; Hamrick et al. 2016), and that they can take actions which help them gather information necessary to make these inferences (Bramley et al. 2018) in accordance with Newtonian laws. It is perhaps then not too surprising at this point that people take simple control actions based on their knowledge of mass too. Having confirmed this is useful, but in light of the prior body of work I am not sure how significant this insight is.

2. While the modeling component of the paper is as a whole impressively precise and rigorous, it does not innovate much beyond past work either, except for the introduction of (1) motor variability and (2) a cost function. The motor variability component is a straightforward addition to existing Bayesian models. The cost function seems interesting, but I was confused about where the form of the cost functions come from. I looked at how they are defined in the appendix, but it still is not clear to me. For example, isn’t the 0-1 cost supposed to be discontinuous? Why then is it represented by a lognormal distribution? Why are there differences in choosing t^int as MAP/mean/median? Perhaps these are standard in the literature on visuomotor control, but this is not my area of expertise and so it would be helpful to readers like myself to derive where these cost functions come from. Moreover, while it is interesting that the quadratic cost function seems to work best, I do not have a good intuition for why this is or what it means, and the paper does not really attempt to provide an intuition. It would be helpful to provide more discussion on what the quadratic cost function implies and what its implications are.

3. I was somewhat disappointed to see that the Bayesian model did not try to capture how people integrate the evidence that they obtain to infer the relevant physical properties. In particular, I felt that one of the most interesting results was that while it seems that people do get some evidence from watching the collisions, this evidence is not as strong as actually getting to see feedback after taking an action. It’s not totally clear to me why this is, and I strongly suspect an ideal Bayesian learner would not capture this difference---especially since it seems within 6 trials of action-feedback people have gotten all the evidence they need, while after 24 trials of watching collisions they still haven’t fully inferred the mass. I think the paper could be much more impactful if it could show not just that people make an inference about mass and use it for control, but how they make these inferences.

Thus, while I think this work merits publishing, I think its potential impact is currently limited. I think that if the paper showed how physical knowledge played into more complex control tasks, or if it could more comprehensively model the entire process of learning, inference, and control, then it would be much stronger. At a minimum, I think the paper needs to do a better job of explaining how the cost functions work.

I also have a few miscellaneous comments:

- Sentences should not start with “e.g.” (lines 40, 45, 72, 76)

- I think Zago & Lacquaniti’s work warrants slightly more discussion than it is given (even if just to say that their work does not focus on inferring unobservable quantities like mass)

- It is probably also worth citing Ullman et al. (2018) and Bramley et al. (2018). The first paper models how people infer unobservable physical quantities and the second paper explores how people select actions to discover information about unobservable physical quantities.

- I did not understand why friction was removed in the collisions in phase 4. Doesn’t this actually make the task harder, since participants would have to jointly infer that friction had changed and infer the mass of the new puck? It would be good if this choice could be further justified.

- I don’t understand what the numbers 1, 2, and 3 are referring to in Figure 2a.

- Lines 236-243: I was initially confused because it seemed to me based on correlation that you could not distinguish between linear vs. square root relationships here. Indeed, it is not until later in the paper that I realized that the way this is distinguished is with the Bayesian model. It would be helpful to mention here that these two hypotheses cannot yet be distinguished but that you will do that later in the modeling section.

- What was the mean absolute error in phase 1, and how does it compare to phase 3?

- Lines 277-279: are these results reversed? “Approximately” is used to describe the results for the light-to-heavy condition, where the p value is listed as <0.001, and for the heavy-to-light condition the results are stated more strongly even though the p value is >0.05.

- Line 299: I am not sure this claim is supported by the evidence presented so far. As far as I can tell, the analyses presented at this point in the paper do not distinguish between linear and square root relationships. Also, I could not find where it was tested that response times scaled linearly with mass at all.

- Equations 3 and 4: should these not have a term for mass in them?

- Line 396: Are these corrected for multiple comparison? If so, please report the correction used. If not, please use a correction here.

- I thought Fig. 19 & 20 were quite interesting that that they should be highlighted more in the main text---they are alluded to in Lines 614-617 but I think they should be referenced more explicitly!

References

Battaglia, P. W., Hamrick, J. B., & Tenenbaum, J. B. (2013). Simulation as an engine of physical scene understanding. Proceedings of the National Academy of Sciences, 110(45), 18327-18332.

Bramley, Gerstenberg, Tenenbaum, and Gureckis (2018). Intuitive experimentation in the physical world. Cognitive Psychology, (195), pp. 9–38

Hamrick, J. B., Battaglia, P. W., Griffiths, T. L., & Tenenbaum, J. B. (2016). Inferring mass in complex scenes by mental simulation. Cognition, 157, 61-76.

Sanborn, A. N., Mansinghka, V. K., & Griffiths, T. L. (2013). Reconciling intuitive physics and Newtonian mechanics for colliding objects. Psychological review, 120(2), 411.

Ullman, Stuhlmüller, Goodman, and Tenenbaum (2018). Learning physical parameters from dynamic scenes. Cognitive Psychology, (104), pp. 57-82

Reviewer #3: This study investigated how people update their actions based on what they learn by observing physical events - in particular, whether new information about a latent variable (mass) is incorporated into behavior in a manner consistent with Newtonian physics. In a clever series of experiments where participants launched pucks attempting to hit a target, the authors found that participants were able to calibrate their launches well in the presence of visual feedback, then carried their assumptions from that experience to their behavior with a new puck of unknown mass, and finally, flexibly updated their behavior on that puck after observing collision events that revealed its mass. Subsequent modeling work showed that the updates to launching behavior were more consistent with Newtonian mechanics than a heuristic model where launch force scales linearly with target distance.

I find the experiments to be well-motivated and well-designed, and the results intriguing, particularly with regard to how readily and precisely participants incorporate their mass inferences into their subsequent behaviors. However, I have a number of concerns about the manuscript in its present form, particularly with regard to the interpretation of the findings as speaking to a link between physical intuitions and actions.

1) A primary goal of this study is to investigate how physical intuitions shape actions. I agree that this is an important and understudied question, but I don't feel that the present study makes a convincing case that the results tell us about action per se. Of course participants must always take some kind of action to report on their decisions, but the action here is an arbitrary one that does not invoke the same movements as actually throwing a puck. Eg, participants didn't need to tense their muscles to an appropriate degree or prepare an appropriate grip aperture as in the lifting studies that were cited. The nature of the action may make a substantive difference - engaging the action system in a naturalistic way may affect the way people access their internal physical models (eg, Schwartz & Black, Inferences through imagined actions). Is there any aspect of the study that can establish that the nature of the action was important? Would the results here be different if, say, participants verbalized a number indicating how hard to launch the puck? If not, then I feel the motivation for the paper needs reworking to focus less on the action per se and more on the updating of beliefs, etc.

2) Here, the distinction between linear and quadratic distance scaling serves as the diagnostic of whether participants are drawing on a Newtonian model, but it certainly seems possible that participants could adopt quadratic scaling in their launching behaviors without really employing any more sophisticated mental model of physics. Participants may learn the quadratic scaling during the feedback phase and then carry that quadratic scaling through their behavior in the rest of the study as a heuristic that worked well when feedback was available. Are there additional ways in which behaviors based on a full-fledged Newtonian model would differ from one that has quadratic scaling but none of the rest of the sophistication of a Newtonian model? Could any additional distinctions be tested here to further make the case for a Newtonian model?

3) For the implicit mass beliefs plotted in Fig. 8 - are the accurate mass beliefs for phase 2 a necessary consequence of the fact that perceptual uncertainty and behavioral variability were estimated for each participant based on the data from that phase? (ie, is there some circularity here?) Or if not - ie, the mass beliefs could have been scattered even though the data was fit in this way - it would be useful to include some intuitive description of how that could have been the case.

minor points:

4) It is hard to know what exactly to take from panel A of fig. 2 - what does the y axis represent? Were there some shots that landed in 4, 5, & beyond that we're not seeing plotted?

5) I don't see where Figs. 2 & 3 are referred to in the text of the results. It would be helpful to reference the figures when describing the corresponding results.

**Have all data underlying the figures and results presented in the manuscript been provided?**

Reviewer #1: No: The Github repo linked in the "data availability" does not have a project associated with this paper. I'm assuming this is accidental, but it would be good to post this pre-publication.

Reviewer #2: No: A URL is given but it is not the full URL yet (I suppose it will be given if the paper is accepted)

Reviewer #3: Yes

PLOS authors have the option to publish the peer review history of their article (what does this mean?). If published, this will include your full peer review and any attached files.

Reviewer #1: Yes: Kevin A Smith

Reviewer #2: No

Reviewer #3: No
---

## [Decision Letter · Decision Letter 1]

24 Aug 2020

Dear Dr. Rothkopf,

We are pleased to inform you that your manuscript 'Intuitive physical reasoning about objects' masses transfers to a visuomotor decision task consistent with Newtonian physics' has been provisionally accepted for publication in PLOS Computational Biology.

-----

As you will see below, the reviewers were pleased with the changes to the manuscript but still had some comments. These are only meant to improve the manuscript and not conditional of acceptance, but we would suggest that you seriously consider them.

Best regards,

Ulrik R. Beierholm

Associate Editor

PLOS Computational Biology

Wolfgang Einhäuser

Deputy Editor

PLOS Computational Biology

Reviewer's Responses to Questions

**Comments to the Authors:**

Reviewer #1: I would like to thank the authors for their thorough response to all of the reviewers. They have adequately addressed all of the comments I had made, as well as any items that the other reviewers had brought to my attention. As such, I would be happy to see this manuscript published, though have some minor comments below related to textual edits and clarity.

* On page 4, lines 143-145, the authors note that participants were told that they could adjust the force, and so acceleration, and so velocity of the puck by holding down the button. Were the participants explicitly told that there would be a linear relationship between the press time and *initial* velocity?

* On page 16, lines 562-564, the authors write about the observed nodes in the observation model, shaded in grey. However, one of the nodes (m^F_j,k) is missing from this description

* Similarly, on lines 574-576 and eq. 6, some of the variables described in the text differ from the way they are displayed in fig. 9 -- e.g., I believe the latent variable v^per_F in the text refers to v^F-P_i in the plot

* On page 3, line 89, it might be more clear to say "do humans adjust their actions to be consistent..."

* On page 4, lines 107-109, it might be clearer to take the clause "and compared it to the prediction of a linear heuristics model" and make it a separate sentence, as it currently sounds like that clause attaches to the participants, instead of describing what the authors did

* On the top of page 14, line 449, I think the sentence is missing a word: "Therefore, in the following [analysis] we used..."

* There are a few cases where there appears to be an extraneous comma that inappropriately splits a clause into two, including:

- line 69: exceptions are studies which...

- line 96: A succession of four phases investigated what...

- line 179: The second experiment clarifies that...

- line 197: The second phase was designed to investigate how...

- line 210: Note that the two...

- line 553: But here we...

- line 631: Importantly, we also tested which of three...

- line 324: a very weak initial hint that...

Reviewer #2: I thank the authors for their extensive comments and revisions following my and the other reviewers' previous reviews. I very much appreciate the new modeling results, and the clarifications regarding the cost functions. I feel overall that my concerns have been sufficiently addressed by the revision, and thus recommend acceptance. I do have some lingering questions/comments but I am confident these can be included in the final version of the paper without too much work. I respond below to the authors’ comments for each of my original points.

My first original concern was with respect to novelty of the findings. In the response, the authors emphasize that the novelty lies in showing that participants can (1) infer mass from visual stimuli and then (2) use the mass to produce an action. This process of producing an action has three important qualities: (2a) it requires computing a non-linear function; (2b) it is continuous; and (less explicitly stated in their response, but I think implied) (2c) it is causal (i.e. it affects the world). I would still argue that some combination of all of these properties has been examined in the prior literature on physical reasoning, though not all at once (and also not incorporating cost function or action variability in the modeling). For example, in Smith & Vul (2013), participants have to continuously move a paddle to catch a ball (2b & 2c). In Battaglia et al. (2013), participants have to continuously adjust a line to predict the direction a tower will fall, conditional on knowing the mass (2a & 2b). In Hamrick et al. (2016), participants infer the mass and then predict if it will fall (1 & 2a). In Bramley et al. (2018) participants take continuous actions to infer things like mass (going in the other direction of inference). So the fact that subsets of these elements exist to some extent in prior works made me (previously) feel that I did not learn very much. But, giving it further consideration, I do believe it is worthwhile to conclusively demonstrate and model their combination, and to incorporate a notion of cost (which I understand much better now; see below).

My second concern was with respect to the novelty of the modeling. In particular, I was confused about the cost functions, and this led me to underestimate the importance of including them in the model. The explanations provided both in the response and in the manuscript itself were very helpful in resolving my confusion; thank you for including them! I understand now that the different cost functions just determine whether people choose the mode, median, or mean of the posterior distribution as their intended action. This could possibly still be stated more explicitly around line 394 in addition to the explanations of how each of these loss functions penalize errors. I also feel that it could be useful to include in the supplemental the derivation of the distributions in Figure 13 as a result of using these cost functions. I realize now they come from the mode/median/mean of the lognormal, but I had to look up what these were, and I don’t know if it’s reasonable to expect readers to know this off the top of their heads. Just including a short derivation would be helpful, such as: “the mean of a lognormal is exp(mu + sigma^2/2), therefore we parametrize the distribution over t^pre with mu=log(t^int)-sigma^2/2 so that the mean corresponds to t^int”. In any case, the results indicate that people choose the mean of the distribution (i.e. they use a quadratic cost function) rather than the mode or median. This is interesting, and the demonstration that the choice of cost function seems to matter so much is, I think, an important contribution to work on physical reasoning, where the cost function has not traditionally been included (as the authors state).

My third concern was with respect to the modeling of the learning process. To address this, the authors included an additional model (the “observation model”) to model the inference of mass given visual observations of collisions in phase 4 of the experiment and prior estimates of mass from phase 2 of the experiment (inferred using the original model, now termed the “interaction model”). These modeling results demonstrate that (1) given more observations, variance in the mass estimates decreases, and that (2) the posterior distribution given observations is broader than the prior distribution (with the justification given that this “stems from the fact that subjects needed to use the uncertain mass beliefs of the red and yellow pucks when observing the collisions and had additional uncertainty stemming from inferring pucks’ velocities”). These are interesting results, and partially answer my questions about why people seemed more uncertain after visual evidence rather than interactive evidence.

I really like how the observation model ties phase 2 and the first part of phase 4 together, and feel it could actually go one step further to tie together all the phases of the experiment. First, I wasn’t sure how p(m^NF) in the observation model is defined, but I assume it’s probably the same prior over mass as in the interaction model. However, it would be really interesting to instead use the estimate inferred in phase 3 using the interaction model. Second, a similar thing could be done to tie the two parts of phase 4 together: use p(mass in phase 4 | observation model) as the prior over mass for the interaction model for the second part of phase 4. If this model can still capture participant’s actions well, I think this would be a really neat and clean result that connects the various parts of the experiment, demonstrating more quantitatively exactly how people are using their knowledge across various phases. Note that I don’t think this is necessary for acceptance, but I do feel it would really strengthen the paper.

Other comments:

The wording in on lines 598-605 was confusing to me and I had to read it several times to understand that this is essentially saying p(mass in phase 4 | observation model) is different from p(mass in phase 4 | interaction model), and then giving hypotheses as to why this is. I think this should be rephrased to be clearer that the two models make different inferences about what the mass is. It would also be helpful to include the green distributions from Figure 18 in Figure 9, as it’s very hard otherwise to understand that these differ. Finally, the reasons given for these differences are hypotheses and should be stated as such as well; I think currently the justifications sound too definitive. I think it is ok to leave confirmation of these hypotheses to future work, though.

Line 387: “Cost functions govern which action, here the press-time, should be chosen given a posterior belief and a cost function, which quantifies how the decision process penalizes errors on the task.” → this statement is awkwardly phrased, I recommend saying something more like “Cost functions govern which action, here the press-time, should be chosen given a posterior belief. Specifically, the cost function quantifies how the decision process penalizes errors on the task.”

Reviewer #3: The authors have thoroughly addressed my original concerns with updates to the manuscript. I have no remaining concerns and feel that this paper provides a valuable contribution to the field.

**Have all data underlying the figures and results presented in the manuscript been provided?**

Reviewer #1: Yes

Reviewer #2: None

Reviewer #3: Yes

PLOS authors have the option to publish the peer review history of their article (what does this mean?). If published, this will include your full peer review and any attached files.

Reviewer #1: **Yes: **Kevin A Smith

Reviewer #2: No

Reviewer #3: No

---

## [Editor Report · Acceptance letter]

9 Oct 2020

PCOMPBIOL-D-20-00210R1 

Intuitive physical reasoning about objects' masses transfers to a visuomotor decision task consistent with Newtonian physics

Dear Dr Rothkopf,

I am pleased to inform you that your manuscript has been formally accepted for publication in PLOS Computational Biology. Your manuscript is now with our production department and you will be notified of the publication date in due course.

With kind regards,

Laura Mallard
